# A comprehensive Bioconductor ecosystem for the design of CRISPR guide RNAs across nucleases and technologies

Luke Hoberecht [1], Pirunthan Perampalam[2], Aaron Lun[1] & Jean-Philippe Fortin [1] ✉

The success of CRISPR-mediated gene perturbation studies is highly dependent on the quality of gRNAs, and several tools have been developed to enable optimal gRNA design. However, these tools are not all adaptable to the latest CRISPR modalities or nucleases, nor do they offer comprehensive annotation methods for advanced CRISPR applications. Here, we present a new ecosystem of R packages, called *crisprVerse*, that enables efficient gRNA design and annotation for a multitude of CRISPR technologies. This includes CRISPR knockout (CRISPRko), CRISPR activation (CRISPRa), CRISPR interference (CRISPRi), CRISPR base editing (CRISPRbe) and CRISPR knockdown (CRISPRkd). The core package, *crisprDesign*, offers a user-friendly and unified interface to add off-target annotations, rich gene and SNP annotations, and on- and off-target activity scores. These functionalities are enabled for any RNA- or DNA-targeting nucleases, including Cas9, Cas12, and Cas13. The *crisprVerse* ecosystem is open-source and deployed through the Bioconductor project (https://github.com/crisprVerse).

The performance of CRISPR-based experiments depends critically on the choice of the guide RNAs (gRNAs) used to guide the CRISPR nuclease to the target site. Variable gRNA on-target activity, as well as unintended off-targeting effects, can lead to inconsistent phenotypic readouts in screening experiments. For the purpose of analyzing pooled screens, many approaches attempt to model gRNA quality in the generation of gene-level scores to improve statistical inference[1–5]. However, suboptimal gRNA design is only partially mitigated by analysis strategies that sacrifice statistical power for robustness to suboptimal guides. One way to increase the signal-to-noise ratio in screening experiments is to enrich gRNA libraries for gRNAs that have high predicted on-target activity. Predicting on-target activity from the spacer sequence is an extensive area of research, and several algorithms leveraging experimental data have been developed for different nucleases and contexts[6–13].

In addition to its sequence, the genomic context of the on- and off-target sites for each gRNA is another important consideration for gRNA design. For example, designing gRNAs that uniquely map to the genome can be challenging, especially for genes sharing high homology with other genomic loci, either in coding or non-coding regions[14]. Furthermore, knowing whether or not an off-target is located in the coding region of another gene can rule out the use of a given gRNA. Finally, genetic variation, such as single-nucleotide polymorphisms (SNPs) and small indels, can have a direct impact on gRNA binding activity and on-target specificity by altering complementarity between spacer sequences and the host cell genomic DNA[15–18].

The rapid increase of CRISPR-based applications and technologies poses another challenge to gRNA library design. A large variety of nucleases are now available and routinely used, including engineered nucleases that recognize a larger set of PAM sequences[19–22] and novel classes of nucleases such as the RNA-targeting Cas13 family[23–25]. Each nuclease comes with its own set of gRNA design rules and constraints. In addition, these nucleases can also be mixed and matched with different types of CRISPR applications, increasing the complexity of

[1]Genentech Research and Early Development, Genentech, Inc., 1 DNA Way, South San Francisco, CA 94080, USA. [2]ProCogia Inc. under contract to Hoffmann-La Roche Limited, Toronto, Canada. ✉e-mail: fortin946@gmail.com

gRNA design. As an example, CRISPR base editing (CRISPRbe)[26,27], which requires additional gRNA design functionalities to capture the editing window and prediction of editing outcomes, can be combined with the Cas13 family to perform RNA editing[28]. Finally, emerging screening modalities, such as optical pooled CRISPR screening[29] and gRNA pairing, require additional specialized gRNA design considerations.

Given the complexity, heterogeneity, and fast growth of the aforementioned CRISPR modalities and applications, it is paramount to develop and maintain adaptable, modular, and robust software for gRNA design. This ensures that the scientific community can efficiently design first-class CRISPR reagents in a timely manner for both well-established and emerging technologies. An ideal gRNA design framework has the following qualities: (1) it offers multiple cutting-edge methods for on-target scoring and off-target prediction based on gRNA sequences, (2) it provides comprehensive gRNA annotation to enable consideration of the genomic context for all gRNA on-target and off-target sites, (3) it already supports (or can be easily extended to) newer CRISPR technologies, including an arbitrary combination of nucleases and modalities, and (4) it easily scales for designing large-scale gRNA libraries for different screening platforms. While a multitude of web applications and command line interfaces has been developed to enable gene- or other target-specific gRNA design[12,30–39], none of the existing tools completely satisfies the requirements listed above.

In this work, we describe a modular ecosystem of R packages, called *crisprVerse*, that enable the design of CRISPR gRNAs across a variety of nucleases, genomes, and applications. The *crisprBowtie* and *crisprBwa* packages provide comprehensive on-target and off-target search for reference genomes, transcriptomes, or any custom sequences. The *crisprScore* package provides a harmonized framework to access a large array of R- and Python-based gRNA scoring algorithms developed by the CRISPR community, for both on-target and off-target scoring. The *crisprBase* package implements functionalities to describe and represent DNA- and RNA-targeting CRISPR nucleases, nickases and base editors, as well as genomic arithmetic rules that are specific to CRISPR design. The package *crisprDesign* provides a user-friendly package to design and annotate gRNAs in one place, including gene and TSS annotation, search for SNP overlap, addition of evolutionary conservation scores, characterization of edited alleles for base editors, sequence-based design rules, and library design functionalities such as ranking and platform-specific considerations. Finally, the package *crisprViz* allows users to visualize gRNAs within genomic tracks, with the option of embedding additional genomic annotations such as SNPs, repeat elements, or chromatin accessibility data. The *crisprVerse* ecosystem currently supports five different CRISPR modalities: CRISPR knockout (CRISPRko), CRISPR activation (CRISPRa),

CRISPR interference (CRISPRi), CRISPR base editing (CRISPRbe) and CRISPR knockdown (CRISPRkd) using Cas13.

We illustrate the rich functionalities of our ecosystem through three case studies: designing gRNAs to edit *BRCA1* using the base editor BE4max, designing gRNAs to knock down *CD55* and *CD46* using CasRx, and designing optimal gRNAs to activate *MMP7* through CRISPRa for different wildtype and engineered nucleases. We also show that our default gRNA ranking criteria yield optimal gRNAs by reanalyzing five genome-wide fitness screening datasets. Our R packages are open-source and deployed through the Bioconductor project[40,41]. This makes our tools fully interoperable with other packages, and facilitates long-term development and maintenance of our ecosystem. Source code, tutorials, and extensive documentation are provided on our website: https://github.com/crisprVerse.

## Results

### *crisprBase* as a core infrastructure package to represent CRISPR nucleases and base editors

The *crisprBase* package implements a common framework in the *crisprVerse* ecosystem for representing and manipulating nucleases and base editors through a set of classes and CRISPR-specific genome arithmetic functions. The *CrisprNuclease* class provides a general representation of a CRISPR nuclease, encoding all of the information necessary to perform gRNA design and other analyses involving CRISPR technologies. This includes the PAM side with respect to the protospacer sequence, recognized PAM sequences with optional tolerance weights, and the relative cut site. Specific *CrisprNuclease* instances can be easily created to represent a diversity of wild-type and engineered CRISPR nucleases (Fig. 1). We also implement a *BaseEditor* subclass that provides additional base editing information such as the editing strand and a matrix of editing probabilities for possible nucleotide substitutions.

### *crisprDesign*: a comprehensive tool to perform complex gRNA design

*crisprDesign* offers a comprehensive suite of methods to design and annotate gRNAs (see Table 1) and represents the core package of the *crisprVerse* ecosystem. For users, the package provides a centralized and streamlined end-to-end workflow for gRNA design, alleviating the burden of using different tools at different stages of the design process. For developers, *crisprDesign* is built on top of a modular package ecosystem that implements the gRNA design tasks (see Table 2 in the Methods section), allowing the same code to be easily re-used outside of CRISPR applications and gRNA design.

Table 1 includes a comparison to ten commonly-used gRNA design softwares: *multicrispr*[38], *CRISPRseek*[39], *CHOPCHOP*[33], *CRISPOR*[34],

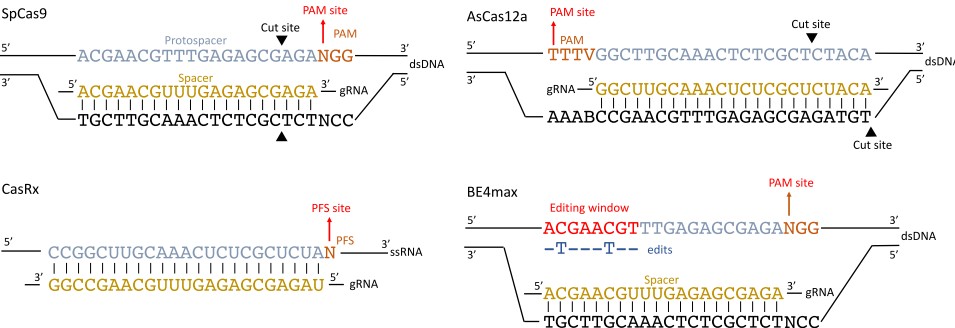

**Fig. 1 | Examples of DNA- and RNA-targeting nucleases represented in *crisprBase*.** gRNA spacer sequences are shown in yellow. Target DNA/RNA protospacer sequences are shown in blue. Protospacer adjacent motifs (PAMs) and protospacer flanking sequences (PFSs) are shown in orange. Nuclease-specific cutting sites are represented by black triangles. For the C to T base editor BE4max, on-target editing happens on the DNA strand containing the protospacer sequence. The editing window varies by base editor. The first nucleotide of the PAM/PFS is used as the representative coordinate of a given target sequence.

**Table 1 | gRNA design functionalities implemented in *crisprVerse* and commonly-used gRNA design tools**

| Category | Functionality | crisprDesign | multicrispr | CRISPRseek | CHOPCHOP | CRISPOR | CCTop | GUIDES | Cas-Designer | FlashFry | E-CRISP | CRISPick |
|---|---|---|---|---|---|---|---|---|---|---|---|---|
| Nuclease | DNA-targeting: Cas9 | ✓ | ✓ | ✓ | ✓ | ✓ | ✓ | ✓ | ✓ | ✓ | ✓ | ✓ |
| | DNA-targeting: Cas12 | ✓ | ✓ | ✓ | ✓ | ✓ | ✓ | ✓ | ✓ | ✓ | ✓ | ✓ |
| | DNA-targeting: Custom | ✓ | Limited | ✓ | Limited | Limited | Limited | | ✓ | Limited | | Limited |
| | RNA-targeting: Cas13 | ✓ | ✓ | ✓ | ✓ | | | | | | | |
| | Nickase | ✓ | ✓ | ✓ | ✓ | | | | | | | |
| Modalities | CRISPRko | ✓ | ✓ | ✓ | ✓ | ✓ | ✓ | ✓ | ✓ | ✓ | ✓ | ✓ |
| | CRISPRbe | ✓ | ✓ | ✓ | ✓ | ✓ | ✓ | | ✓ | | | |
| | CRISPRa | ✓ | | | ✓ | | | | | ✓ | ✓ | ✓ |
| | CRISPRi | ✓ | | | ✓ | | | | | ✓ | ✓ | ✓ |
| | RNA editing (CRISPRkd) | ✓ | | | ✓ | | | | | | | |
| | Optical Pooled Screening | ✓ | | | | | | | | | | |
| | Prime editing | * | ✓ | ✓ | | | | | ✓ | | | † |
| Target space | Reference genomes | ✓ | ✓ | ✓ | ✓ | ✓ | ✓ | Limited | ✓ | ✓ | ✓ | ✓ |
| | Custom sequences | ✓ | ✓ | ✓ | ✓ | ✓ | ✓ | | ✓ | ✓ | ✓ | ✓ |
| Off-target aligner | Bowtie | ✓ | ✓ | ✓ | ✓ | ✓ | ✓ | ✓ | | | ✓ | |
| | BWA | ✓ | | | | ✓ | | | | | | |
| | Other | Biostrings | Biostrings | Biostrings | | | | | Cas-OFFinder | FlashFry | | |
| Off-target options | Genomic coordinates | ✓ | ✓ | ✓ | ✓ | ✓ | ✓ | | ✓ | ✓ | | |
| | Custom sequences | ✓ | | ✓ | ✓ | | | | | | | |
| | Cross-reactivity | ✓ | | | | | | | | | | |
| | Minor/major alleles | ✓ | | | | | | | | | | |
| On-target scoring | Rule Set 1 | ✓ | ✓ | ✓ | ✓ | ✓ | | | | ✓ | ✓ | ✓ |
| | Azimuth | ✓ | ✓ | ✓ | ✓ | ✓ | | ✓ | | ✓ | | ✓ |
| | Rule Set 3 | ✓ | | | ✓ | | | | | | | |
| | CRISPRscan | ✓ | | ✓ | ✓ | ✓ | | | | ✓ | | |
| | CRISPRater | ✓ | | | | ✓ | ✓ | | | | | |
| | DeepCpf1 | ✓ | ✓ | ✓ | ✓ | ✓ | | | | | | ✓ |
| | DeepSpCas9 | ✓ | | | | | | | | | | |
| | DeepHF | ✓ | | | | | | | | | | |
| | Lindel | ✓ | | ✓ | ✓ | ✓ | | | | | | |
| | CRISPRai | ✓ | | | | | | | | | | ✓ |
| | EnPAM+GB | ✓ | | | | | | | | | ✓ | |
| | CasRx-RF | ✓ | | | | | | | | | | |
| | PAM scoring | ✓ | | | | | | | | | | |
| Off-target scoring | MIT | ✓ | ✓ | ✓ | ✓ | ✓ | | | | ✓ | | ✓ |
| | CFD | ✓ | ✓ | ✓ | ✓ | ✓ | | ✓ | | ✓ | | ✓ |
| | CasRx | ✓ | | ✓ | ✓ | | ✓ | | | | | |
| Annotations | Off-target annotation | ✓ | ✓ | | | ✓ | | | | | | ✓ |
| | Isoform specification | ✓ | | | | | | | | | ✓ | |
| | Reinitiation sites | ✓ | | ✓ | ✓ | | | | | | | ✓ |

**Table 1 (continued) | gRNA design functionalities implemented in crisprVerse and commonly-used gRNA design tools**

| | crisprDesign | multicrispr | CRISPRseek | CHOPCHOP | CRISPOR | CCTop | GUIDES | Cas-Designer | FlashFry | E-CRISP | CRISPick |
|---|---|---|---|---|---|---|---|---|---|---|---|
| Pfam domains | ✓ | | | | | | | | | | |
| SNP annotation | ✓ | | | | ✓ | | | | | | |
| TSS annotation | ✓ | | | | | | ✓ | | | ✓ | ✓ |
| Conservation | ✓ | | | | | | | | | | |
| Restriction sites | ✓ | | ✓ | ✓ | ✓ | ✓ | | | | ✓ | |
| PolyT signal | ✓ | | ✓ | | ✓ | | | | | ✓ | |
| GC content | ✓ | | ✓ | ✓ | ✓ | | | ✓ | | ✓ | |
| Hairpin loops | ✓ | | | ✓ | | | | | ✓ | | |
| Paired gRNAs | ✓ | ✓ | ✓ | ✓ | | | | | | ✓ | |
| Ranking | ✓ | ✓ | | ✓ | | ✓ | | | ✓ | | ✓ |

Check marks indicate which functionalities are present in each tool at time of publication. * In progress. † Information could not be found. See the Methods section for a detailed description of the criteria used for assessing feature availability.

**Table 2 | R packages in the crisprVerse ecosystem**

| R package | Description |
|---|---|
| crisprVerse | Easy install of the crisprVerse ecosystem |
| crisprDesign | Core package for gRNA design |
| crisprBase | Nuclease specification and gRNA arithmetics |
| crisprBowtie | gRNA spacer alignment with *Bowtie* |
| crisprBwa | gRNA spacer alignment with *BWA* |
| crisprScore | On- and off-target scoring algorithms for gRNAs |
| crisprViz | Visualization of gRNAs using genomic tracks |
| Rbwa | R wrapper for *BWA* aligner |
| crisprScoreData | Pre-trained machine learning models for *crisprScore* |
| crisprDesignData | Pre-computed data for the crisprVerse ecosystem |

*CCTop*[35], *GUIDES*[42], *Cas-Designer*[43], *FlashFry*[36], *E-CRISP*[30] and CRISPick; see the Methods section for a detailed description of the criteria used for benchmarking. While several of the features implemented in *crisprDesign* are also available in other tools, *crisprDesign* provides the most complete gRNA design solution across nucleases and modalities. Unlike *crisprDesign*, many of the other tools do not provide informative on- and off-target annotations, limiting their use for optimal gRNA selection. In the following sections, we describe each of the gRNA design components and functionalities that are available in *crisprDesign*.

### Representation of gRNAs using the *GuideSet* container

The genomic coordinates of gRNA protospacer sequences in a target genome can be represented using genomic ranges. The Bioconductor project[40,41] provides a robust and well-developed core data structure, called *GRanges*[44]), to efficiently represent genomic intervals. We provide in *crisprDesign* an extension of the *GRanges* class to represent and annotate gRNA sequences: the *GuideSet* container. Briefly, the container extends the *GRanges* object to store additional project-specific metadata information, such as the CRISPR nuclease employed and target mRNA or DNA sequences (if different from a reference genome), as well as rich gRNA-level annotation columns such as on- and off-target alignments tables and gene context annotations. In Fig. 2, we show an example of a *GuideSet* storing information about gRNAs targeting the coding sequence of *KRAS* using SpCas9.

### *crisprScore* implements state-of-the-art scoring methods

Predicting on-target binding and cutting efficiency of gRNAs is an extensive area of research. Many algorithms have been developed to tackle this problem, basing their prediction on a variety of features: sequence composition of the spacer sequence and flanking regions, including nucleotide content and melting temperature, cell type-specific chromatin accessibility data, and distance to transcription starting site (TSS). Unfortunately, the heterogeneity in the algorithm implementations hinders the practical use of those algorithms: some methods are implemented in Python 2[8,10,13,45], in Python 3[9,22,46], or in R[7,12,47,48]. In addition, the required inputs, data structures, and terminology are not consistent across software and algorithms, increasing the likelihood of user error. Finally, several of the algorithms are currently not bundled up into easy-to-use packages, limiting their accessibility and therefore their usage.

To resolve this, we created a general and harmonized framework for on-target and off-target prediction of gRNAs, implemented in our R package *crisprScore*. The philosophy behind *crisprScore* is to abstract away from the user the language, implementation, and complexity of the different algorithms used for prediction. It uses the Bioconductor package *basilisk*[49] to seamlessly integrate and manage incompatible Python modules in one user session. This enables *crisprScore* to centralize all Python-based scoring algorithms together with R-based

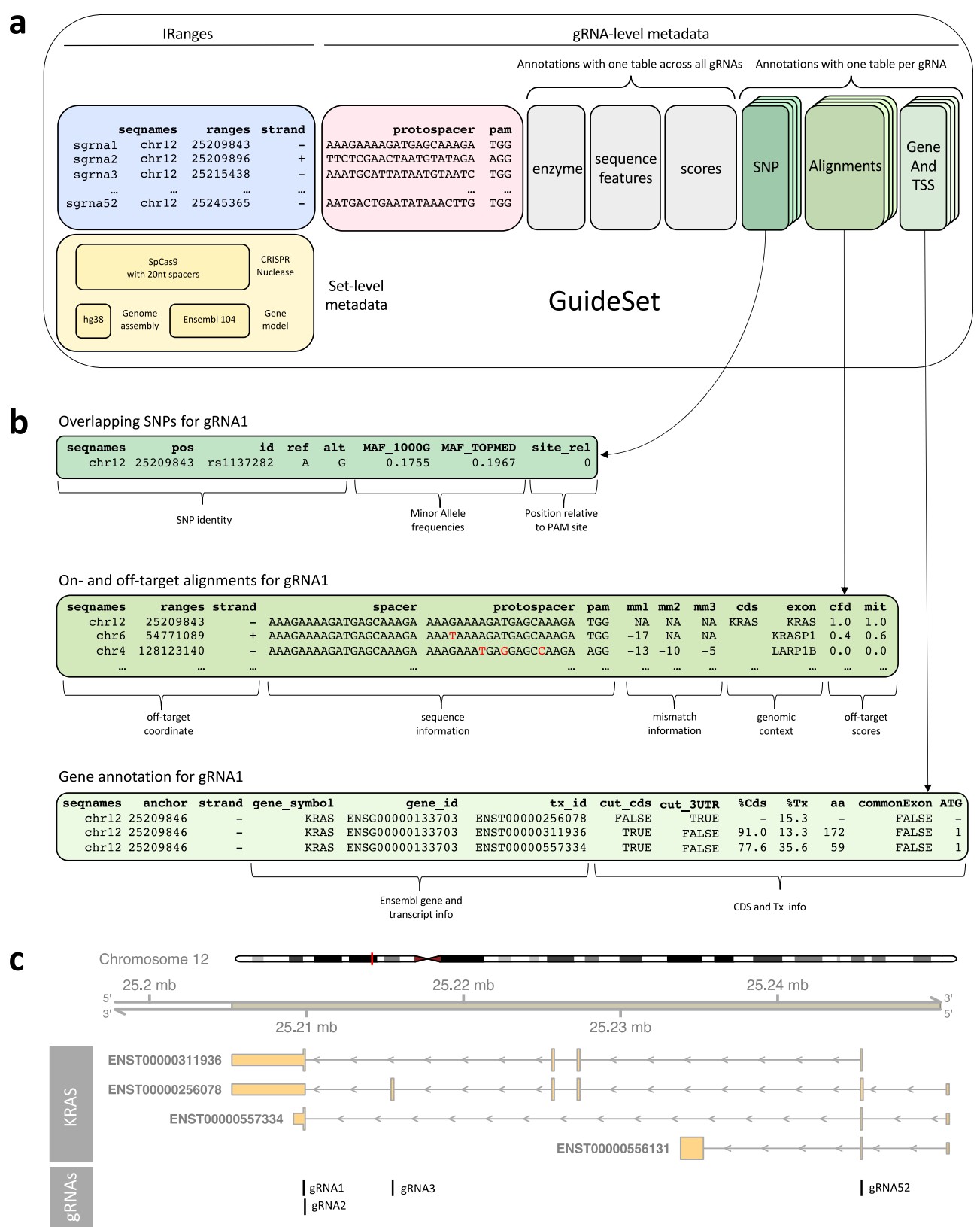

prediction algorithms, reporting all scores in a single data frame for convenient inspection. By providing a harmonized user interface, our framework facilitates methods comparison.

We note that while the package provides a harmonized framework from a user perspective, it also allows each scoring algorithm to be implemented with its own sets of parameters and inputs. We have included as many methods as possible (Table 3), with the goal of democratizing the use of different scoring algorithms in an unbiased manner. Developers can easily contribute new methods to the *crispr-Score* package as they become available.

**Fig. 2 | Example of a GuideSet container for gRNAs targeting _KRAS_ using SpCas9. a** The blue box stores the genomic coordinates in GRCh38 to represent the target protospacer sequences using a GRanges object. By convention, we use the first nucleotide of the PAM sequence (in the 5′ to 3′ direction) as the representative genomic coordinate of protospacer sequences. The pink box stores sequence information of the protospacers and PAMs. The yellow box represents global metadata used for creating the GuideSet, including a formal _CrisprNuclease_ object, the reference genome of the target protospacers, and gene model used for annotation. The gray boxes are examples of optional gRNA-level metadata columns that store information about enzyme restriction sites, spacer sequence features such as GC content, and on- and off-target scores. The green boxes represent optional per-gRNA annotations for SNP overlap, on- and off-target alignments, and gene context; each annotation stores a detailed table (2) dimensions) for each gRNA (3rd dimension). **b** Selected annotations for gRNA 1 corresponding to the row highlighted in the green boxes of (**a**). **c** The first genomic track represents the four annotated protein-coding isoforms of human gene _KRAS_ in GRCh38 coordinates. The second track shows the 4 gRNAs shown in the blue box of (**a**).

## Table 3 | On-target and off-target scoring methods currently available in _crisprScore_

| Nuclease | Variant | Method | Type | Reference |
|----------|---------|--------|------|-----------|
| SpCas9 | WT | Rule Set 1 | On-target efficiency | [7] |
| | WT | Azimuth | On-target efficiency | [8] |
| | WT | Rule Set 3 | On-target efficiency | [74] |
| | WT | CRISPRscan | On-target efficiency | [12] |
| | WT | CRISPRai | On-target efficiency | [13] |
| | WT | DeepHF | On-target efficiency | [9] |
| | HiFi | DeepHF | On-target efficiency | [9] |
| | WT | Lindel | On-target efficiency | [46] |
| | WT | DeepSpCas9 | On-target efficiency | [45] |
| | WT | CRISPRater | On-target efficiency | [48] |
| | WT | MIT | Off-target cutting | [57] |
| | WT | CFD | Off-target cutting | [6] |
| AsCas12a | WT | DeepCpf1 | On-target efficiency | [10] |
| | Enhanced | enPAM+GB | On-target efficiency | [22] |
| RfxCas13d | WT | CasRx-RF | On-target efficiency | [47] |
| | WT | CasRx-CFD | Off-target cutting | [104] |

### _crisprDesign_ enables fast characterization and annotation of off-targets

Off-targeting effects can occur when a spacer sequence maps with perfect or imperfect complementarity to a genomic locus other than the primary on-target. Given that nucleases can still bind and cut in the presence of nucleotide mismatches between spacer sequences and target DNA sequences[50–52], it is paramount to obtain and characterize all putative mismatch-mediated off-targets.

The off-target functionalities in _crisprDesign_ are divided into two parts: off-target search (alignment) and off-target characterization (genomic context and scoring). For the off-target search, we offer three different alignment methods: _Bowtie_[53], the BWA-backtrack algorithm in _BWA_[54] and the Aho-Corasick exact string matching method implemented in _Biostrings_[55,56]. We developed two independent R packages to implement the _Bowtie_ and _BWA_ alignment methods: _crisprBowtie_ and _crisprBwa_. Notably, the packages were developed to work with any nucleases, and for both DNA and RNA target spaces (reference genomes and transcriptomes, respectively). While the maximum number of mismatches for _Bowtie_ is limited to 3, there is no limit for _BWA_.

Given the short nature of gRNA spacer sequences, both _Bowtie_ and _BWA_ are ideal tools for off-target search and provide ultrafast results. On the other hand, the alignment method based on the Bioconductor package _Biostrings_ does not need the creation of a genome index, and is particularly useful for off-target search in short custom sequences. All methods can be invoked via the _addSpacerAlignments_ function, which returns the on- and off-target alignments as a _GRanges_ object in the _GuideSet_ metadata.

To add genomic context to the on- and off-targets, a _TxDb_ object can be provided to the _addSpacerAlignments_ function. The _TxDb_ object is a standard Bioconductor object to store information about a gene model, and can easily be made from transcript annotations available as GFF3 or GTF files. Gene annotation columns are added to the off-target table for different contexts: 5′ UTRs, 3′ UTRs, CDS, exons, and introns. Finally, users can add the MIT and CDF off-target specificity scores[8,57] implemented in _crisprScore_ to characterize the likelihood of nuclease cleavage at the off-targets.

### Comparison of the off-target alignment methods

We first compared the accuracy of the three alignment methods implemented in _crisprDesign_. It was previously reported using 10 spacer sequences that _Bowtie_ misses a large number of double-mismatch and triple-mismatch off-targets in comparison to the gold-standard complete string matching algorithm[38]. To investigate this, we repeated the PAM-agnostic on- and off-target alignment of the 10 spacer sequences to the GRCh38 reference genome using all three alignment methods. All three alignment methods implemented in _crisprDesign_ return an identical list of off-targets (see Supplementary Table 1). This indicates that, contrary to previous reports, both _BWA_ and _Bowtie_ provide a complete on- and off-target search. It appears that the previously-reported missing off-targets are located on unlocalized and unplaced GRCh38 sequences.

Next, we evaluated the run times of four configurations offered in _crisprDesign_ for alignment: _Bowtie_, _BWA_, an iterative version of _Bowtie_ (bowtie-int) and an iterative version of _BWA_ (bwa-int). We developed iterative versions of the _Bowtie_ and _BWA_ alignments to avoid situations where gRNAs are mapping to hundreds of loci in the genome, considerably slowing down the off-target search when a higher number of mismatches is allowed. The iterative strategy starts by aligning spacer sequences with no mismatches allowed. Then, it sequentially performs the alignment with a higher number of mismatches only for sequences that have a low number of off-targets at the previous step, thus avoiding the cost of extra searches for low-quality sequences that already have many off-targets. We performed the evaluation on three sets of gRNAs targeting the human genome, each with a different size (see Methods). For all three sets, the _Bowtie_ and BWA gRNA alignments have comparable run times. (Supplementary Fig. 1). For _ZNF101_, which contains several non-specific gRNAs overlapping a repeat element, our iterative versions of the alignment methods shows substantial gain in speed.

Finally, we compared run times for designing SpCas9 gRNAs and performing a genome-wide off-target search for the following tools: _CCTop_, _CHOPCHOP_, _multicrispr_, _FlashFry_, and _crisprDesign_. Other tools were not included for reasons discussed in the Methods section. To perform the evaluation, we generated six random subsets of protein-coding exons located on chromosome 1 with the following sizes: 100, 200, 400, 800, 1600, and 3200 exons. For each tool and each subset, we ran the off-target alignment against the human reference genome (GRCh38 build) using a maximum of 2 mismatches. We included the alignment parameters used for each tool in the Methods section. Both _FlashFry_ and the iterative _Bowtie_ alignment implemented in _crisprDesign_ show a substantial speed gain in comparison to other methods (Supplementary Fig. 2).

## Accounting for human genetic variation by adding SNP annotation

Genetic variation, such as SNPs and small indels, can have a direct impact on gRNA binding productivity and on-target specificity by altering complementarity between spacer sequences and the target DNA or RNA[15–18]. In *crisprDesign*, users can apply the function *addSNPAnnotation* to annotate gRNAs for which the target protospacer sequence overlaps a SNP. This enables users to discard or flag undesirable gRNAs that are likely to have variable activity across different human genomes.

Given that the current human reference genome was built using only a small number of individuals, the allele represented in the human reference genome at a particular locus does not always correspond to the major allele in a population of interest. Inspired by the major-allele reference genome indices provided by the *Bowtie* team (see https://github.com/BenLangmead/bowtie-majref), we created two new human genomes to be used throughout our ecosystem that represent the major allele and the minor allele using dbSNP151 (see Methods). Both genomes are available in Bioconductor as *BSgenome* packages. Both packages can be used in our ecosystem to improve gRNA design by designing gRNAs against either the minor or major allele genome, and searching for off-targets in both the major and minor allele genomes.

## Comprehensive gene and functional annotations

The genomic context of the on-target sites is paramount for optimally selecting gRNAs in most, if not all, CRISPR applications. *crisprDesign* includes the *addGeneAnnotation* and *addTssAnnotation* functions, which report comprehensive transcript- and promoter-specific context for each gRNA target site, respectively. Users simply need to provide a standard Bioconductor *TxDb* object to specify which gene model should be used to annotate on- and off-targets.

For CRISPRko applications, *addGeneAnnotation* annotates which isoforms of a given gene are targeted. It also adds spatial information about the relative cut site within the coding sequence of each isoform, which has been shown to contribute to gRNA activity[8]. Since translation reinitiation can result in residual protein expression[58], *addGeneAnnotation* reports whether or not the gRNA cut site precedes any downstream in-frame ATG sequences using a set of published rules[59]. Additionally, to maximize gene knockout based on protein domains[60], we include Pfam domain annotation[61] via the *biomaRt* package[62]. For CRISPRa and CRISPRi applications, *addTssAnnotation* indicates which promoter regions are targeted by each gRNA, as well as the location of the target cut site relative to the TSS. This allows the user to easily select guides in the optimal targeting window.

To further put the gRNA targets into biological context, users can access thousands of genomic annotation datasets through the Bioconductor *AnnotationHub* resource. The resource includes common sources such as Ensembl, ENCODE, dbSNP and UCSC. Where appropriate, those annotations are in the *GenomicRanges* format, which make them directly compatible with the *GuideSet* object used to represent gRNAs in our ecosystem. By leveraging overlap operations on *GenomicRanges*, users can identify which gRNAs are present or absent in a given set of annotated features by using a few lines of code. For example, users can ask *AnnotationHub* whether a gRNA is targeting repeat elements to avoid cutting-induced toxicity, or whether a gRNA targets the region upstream of an annotated Cap Analysis of Gene Expression (CAGE) peak for CRISPRa applications. Additionally, the *rtracklayer* Bioconductor package[63] provides functionalities to easily read genome annotations that are stored in the commonly-used WIG, BED, bigWig and bedGraph formats. Utilizing *rtracklayer*, *crisprDesign* provides the function *addConservationScores* to annotate gRNAs with evolutionary conservation scores obtained from the UCSC genome browser (see Methods).

## Advanced functionalities for designing screening libraries

Efficient cleavage can be disrupted by certain features of the gRNA sequence, such as very low or high percent GC content[7,64,65], homopolymers of four nucleotides or longer[66,67], and self-complementarity conducive to hairpin formation[68,69]. When gRNAs are expressed from a U6 promoter, thymine homopolymers (TTTT) are particularly undesirable as they signal transcription termination. The *addSequenceFeatures* function flags all gRNAs that contain such undesirable sequence features. Another consideration in designing gRNA libraries is to exclude spacer sequences that are not compatible with the oligonucleotide cloning strategy. gRNAs that contain restriction sites of the enzymes used to clone the spacer sequences into a lentiviral vector should be excluded. The *addRestrictionEnzymes* function flags all gRNAs that contain restriction enzyme recognition motifs.

Optical pooled screening (OPS) is a promising screening modality that combines image-based in situ sequencing of gRNAs and optical phenotyping on the same physical wells[29]. This enables linking genomic perturbations with high-content imaging at large scale. In such experiments, gRNA spacer sequences are partially sequenced. This translates to additional gRNA design constraints to ensure sufficient dissimilarity of the truncated spacer sequences. *crisprDesign* contains a suite of design functions that take into account OPS constraints, while ensuring that the final OPS library is enriched for gRNAs with best predicted activity.

To assist with the design of complex libraries, we developed the package *crisprViz* to visualize gRNAs. The package uses the Bioconductor package *Gviz*[70] to offer a flexible and integrated visualization of gRNAs along genomic coordinates. Users can visually inspect gRNAs within a genomic track with the option of adding annotation tracks such as transcript models, SNP annotations, repeat elements, and nucleotide sequences.

## Functional annotations in *crisprDesign* improve gRNA selection

We illustrate how functional annotations implemented in *crisprDesign* can improve gRNA selection by focusing on two functionalities: *addConservationScores* and *addGeneAnnotation*. We assessed the *addConservationScores* function using the large-scale CRISPRko fitness screening dataset from Project Achilles[1]. We obtained normalized log fold changes (LFCs) measuring gRNA dropout over time (see Methods). In fitness screens, gRNAs targeting essential genes should deplete over time, and are therefore expected to have negative LFCs. Therefore, gRNAs targeting essential genes can be used to investigate determinants of gRNA activity. We downloaded basewise phyloP scores[71] from the UCSC genome browser. Scores were calculated from a phylogenic alignment of 30 vertebrate species (see Methods). Positive and negative scores represent evolutionary conservation and acceleration, respectively. In Supplementary Fig. 3a, we show the correlation between LFCs and conservation scores obtained using different window sizes, for gRNAs targeting a reference set of essential genes[72]. The data suggest an optimal window of 18 nucleotides around the cut site, which is our recommended window size in *crisprDesign*. In Supplementary Fig. 3b, we present LFC distributions of gRNAs targeting essential genes, split by the sign of the gRNA conservation score. gRNAs targeting conserved regions (positive score, red line) show greater activity than less conserved regions (negative score, black line) as observed by greater gRNA dropout. This is in line with previous results[67,73,74]. gRNAs targeting non-essential genes serve as negative controls and show no dropout, irrespective of the conservation scores, as expected (Supplementary Fig. 3c).

Next, we sought to understand how gRNA position within the CDS of the target gene impacts gRNA activity. Given that most gRNAs in Project Achilles were located in the first 50% of the target CDS by design, we obtained a different screening dataset; we downloaded data from a genome-wide fitness screen performed in HCT116 cells (Hart2015 dataset, see Methods). We used the *addGeneAnnotation*

**Table 4 | Genome-wide human CRISPRko screen datasets used for comparing SpCas9 gRNA rankings**

| Dataset | gRNA library | Cell Line | Number of gRNAs | Reference |
|---|---|---|---|---|
| Achilles | Avana | (many) | 67,816 | [1] |
| Hart2015 | TKOv1 | HCT116 | 164,576 | [99] |
| Hart2017 | TKOv3 | HAP1 | 81,967 | [100] |
| Wang2015 | Sabatini | K562 | 166, 855 | [101] |
| Tzelepis2016 | Yusa | HL60 | 85,192 | [102] |

function in *crisprDesign* to annotate gRNAs with a position percentage of the target CDS. We used the Ensembl canonical transcript of the target gene as the representative CDS. In Supplementary Fig. 3d, we show the relationship between LFCs and % CDS for gRNAs targeting essential genes. gRNAs located beyond the first 85% of the CDS (to the right of the vertical line) show a progressive decline in activity. The results agree with the litterature[8]. gRNAs targeting non-essential genes serve as negative controls and behave as expected (Supplementary Fig. 3e).

Based on these results, both functional annotations help selecting more active gRNAs; we recommend in *crisprDesign* to prioritize gRNAs with positive conservation scores, and located in the first 85% of the target CDS. Those recommended parameters are implemented as the default parameters in the *crisprDesign* gRNA ranking procedure discussed next.

## gRNA ranking from crisprDesign returns optimized gRNAs

To complement gRNA annotation and assist in library design, *crisprDesign* provides a gRNA ranking function called *rankSpacers*. The function implements our recommended ranking parameters for the nucleases SpCas9, enCas12a, and CasRx, effectively enabling library design automation across targets. It is designed to optimize both on-target activity and minimize off-targeting effects, and includes the functional annotations described in the previous section. Details are provided in the Methods section.

We compared our default gRNA ranking procedure to other tools listed in Table 1 that provide gRNA rankings: *CHOPCHOP*, *CCTop*, *FlashFry* and *CRISPick*. To perform the evaluation, we designed and ranked SpCas9 gRNAs for all human protein-coding genes (Ensembl release 104) using each tool separately (see Methods). Next, we obtained and processed 5 human genome-wide fitness screen datasets from published studies (Table 4), each performed using a different gRNA library. For each dataset and gRNA, a LFC between later and earlier time point samples was calculated to quantify gRNA dropout over time.

gRNAs targeting essential genes are expected to drop out and can be used for benchmarking purposes. To investigate the relationship between gRNA activity and gRNA ranking, we considered for each gRNA library the subset of gRNAs targeting a common reference set of essential genes[72]. For each gene and tool, we identified the top 15 ranked gRNAs based on the tool-specific in silico ranking. In Fig. 3a, we show the distributions of LFCs in the Hart2015 dataset based on two groups: red lines show the distributions of the top 15 ranked gRNAs across genes, and green lines show the distributions of remaining gRNAs. Top ranked gRNAs from *CRISPick* and *crisprDesign* show greater activity than lower ranked gRNAs, as indicated by a negative shift in the red distributions with respect to the green distributions.

We repeated the analysis for each dataset, and summarized the performance of the top ranked gRNAs at discriminating active gRNAs by calculating the difference in means between the green and red distributions (Δ LFC). Results are shown in Fig. 3b. Higher Δ LFCs indicate better performance, and results indicate that both *CRISPick* and *crisprDesign* perform well across all datasets.

## Case study 1: Designing gRNAs targeting *BRCA1* for the base editor BE4max

CRISPR base editors are deaminases fused to CRISPR nickases to introduce mutations at loci targeted by the gRNAs without introducing double-stranded breaks (DSBs)[26,27]. A recent study showed high heterogeneity and complexity of the editing outcomes across eight popular base editors, motivating the need of robust but flexible software to design gRNAs for base editing applications[75]. In particular, this includes functionalities for listing and characterizing potential edited alleles introduced by gRNAs to inform the phenotypic readouts created by those gRNAs.

To illustrate the functionalities of our ecosystem for designing base editor gRNAs, we designed and characterized all possible gRNAs targeting the coding sequence of *BRCA1* for the cytidine base editor BE4max[76]. The design workflow is shown in Fig. 4.

The first step consisted of designing all possible guides targeting *BRCA1* using the *findSpacers* function in *crisprDesign*. The BE4max *BaseEditor* object from *crisprBase* was used to store nucleotide- and position-specific editing probabilities (see Fig. 4a), which inform the editing window of interest for each of the gRNA targets. Next, using the function *getEditedAlleles*, we generated and stored all possible editing events at each gRNA (see Fig. 4b). The function also adds a score for each edited allele that quantifies the likelihood of editing to occur based on the editing probabilities stored in the *BaseEditor* object (see Methods). In addition, each edited allele is annotated for its predicted functional consequence: silent, missense, or nonsense mutation. In case several mutations occur in a given edited allele, the most consequential mutation is used to label the allele (nonsense over missense, and missense over silent). For each gRNA, and for each mutation type, we then generated a gRNA-level score by aggregating the likelihood scores across all possible alleles (see Methods). The score represents the likelihood of a gRNA to induce a given mutation type (see Fig. 4c, left plot).

To show how our gRNA annotations can be used to understand the phenotypic effects observed in screening data, we obtained data from a negative selection pooled screen performed in MelJuSo using a base editing library tiling the *BRCA1* gene[77]. Given that loss-of-function mutations in *BRCA1* reduce cell fitness[78], gRNAs introducing nonsense mutations are expected to drop out. We created Receiver Operating Characteristic (ROC) curves to measure how well gRNA dropout can separate positive controls from other gRNAs. We used LFCs in gRNA abundance between the later time point and the plasmid DNA (pDNA) library as a measure of gRNA dropout (see Methods). We used several thresholds of the nonsense mutation score to label gRNAs as positive controls or not. We observed that gRNA dropout in the screen can separate positive controls well from all other gRNAs, and that performance is improved when using positive controls defined by higher nonsense mutation scores (Fig. 4c).

We also characterized gRNAs for off-targeting effects using *crisprBowtie*, added sequence features using *crisprDesign*, and added on-target scores using *crisprScore*. We asked whether or not the magnitude of gRNA dropout in the screen associates with predicted on-target activity for the SpCas9 nuclease. In Fig. 4d, we show gRNA dropout as a function of different predicted gRNA efficacy scores: Rule Set 1, Azimuth, and DeepHF. gRNAs predicted to induce nonsense mutations are shown in red, and gray otherwise. Despite the fact that each algorithm was trained on data using a SpCas9 nuclease with intact endonuclease activity, gRNA dropout and predicted gRNA efficacy correlate for all methods ($r = -0.30$ for Rule Set 1, $r = -0.20$ for Azimuth, and $r = -0.17$ for DeepHF). Overall, the different functionalities implemented in our ecosystem provides a set of informative annotations for base editor gRNAs and facilitate the interpretation of experimental data obtained from base editor screens.

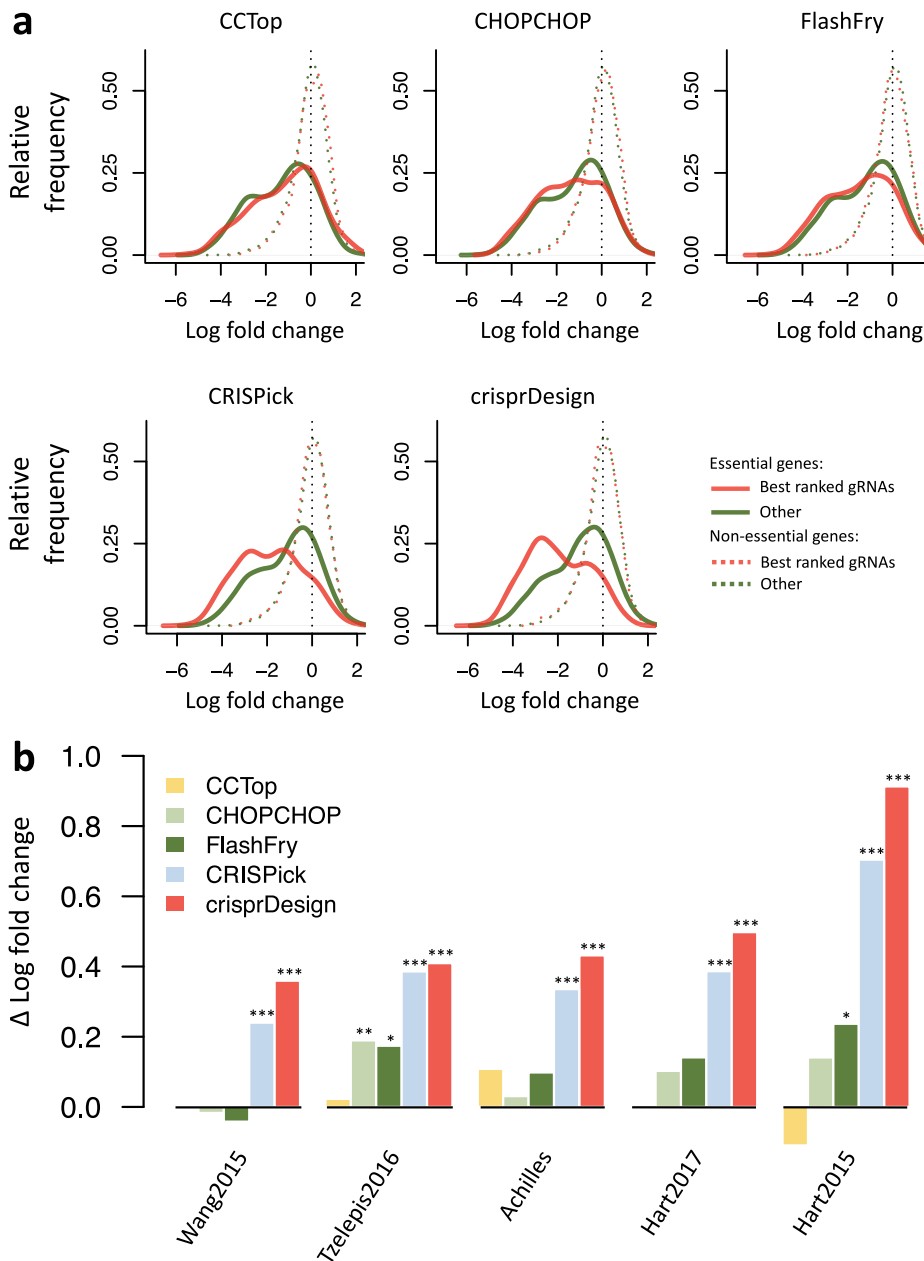

**Fig. 3 | Comparison of CRISPRko Cas9 gRNA rankings for protein-coding human genes.** We designed and ranked gRNAs targeting all protein-coding human genes (Ensembl release 104) using tools that provide gRNA rankings: *CCTop*, *CHOPCHOP*, *FlashFry*, *CRISPick* and *crisprDesign*. To compare gRNA ranking performance across tools, we obtained gRNA LFCs from 5 genome-wide CRISPRko fitness screening datasets, listed in Table 3. In these fitness screens, active gRNAs targeting essential genes are expected to drop out and show negative LFCs. To investigate the relationship between gRNA activity and gRNA ranking, we considered for each gRNA library the subset of gRNAs targeting a common reference set of essential genes[72]. For each gene and tool, we identified the top 15 ranked gRNAs based on the tool-specific in silico ranking. **a** LFC distributions in the Hart2015 dataset for gRNAs targeting essential genes (solid lines) and gRNAs targeting non-essential genes (dotted lines). Red lines show the distributions of the top 15 ranked gRNAs across genes, and green lines show the distributions of remaining gRNAs. For essential genes, top ranked gRNAs from *CRISPick* and *crisprDesign* show greater activity than lower ranked gRNAs (red distributions are negatively skewed). As expected, there are no differences for gRNAs targeting non-essential genes. **b** We repeated the analysis described in (**a**) for each dataset. We summarized the performance of the top ranked gRNAs by calculating the difference in means between the green and red distributions (Δ LFC), for essential genes only. A higher Δ LFC indicates better performance. For each method and dataset, a two-sided t-test was performed to quantify the difference in LFCs between the top ranked gRNAs and the remaining gRNAs. Corresponding p-values are reported above the bars (*p-value < 0.05; **p-value < 0.01; ***p-value < 0.001). Exact p-values are provided in the source data. Source data are provided as a Source Data file.

## Case study 2: Annotating and scoring gRNAs for gene knock-down using CasRx

One of the challenges in designing gRNAs specific to RNA-targeting nucleases is to enable on-target and off-target characterization to be performed in a transcriptome space, as opposed to a reference genome. This requires strand-specific functionalities, transcriptome-specific alignment indexes, as well as additional gene annotation functionalities to capture isoform-specific targeting.

Here, we describe a workflow for designing gRNAs targeting *CD46* and *CD55* using the RNA-targeting nuclease CasRx (RfxCas13d)[25] (Fig. 5). The workflow takes into consideration the aforementioned issues. To validate our design process, we obtained CasRx pooled screening data

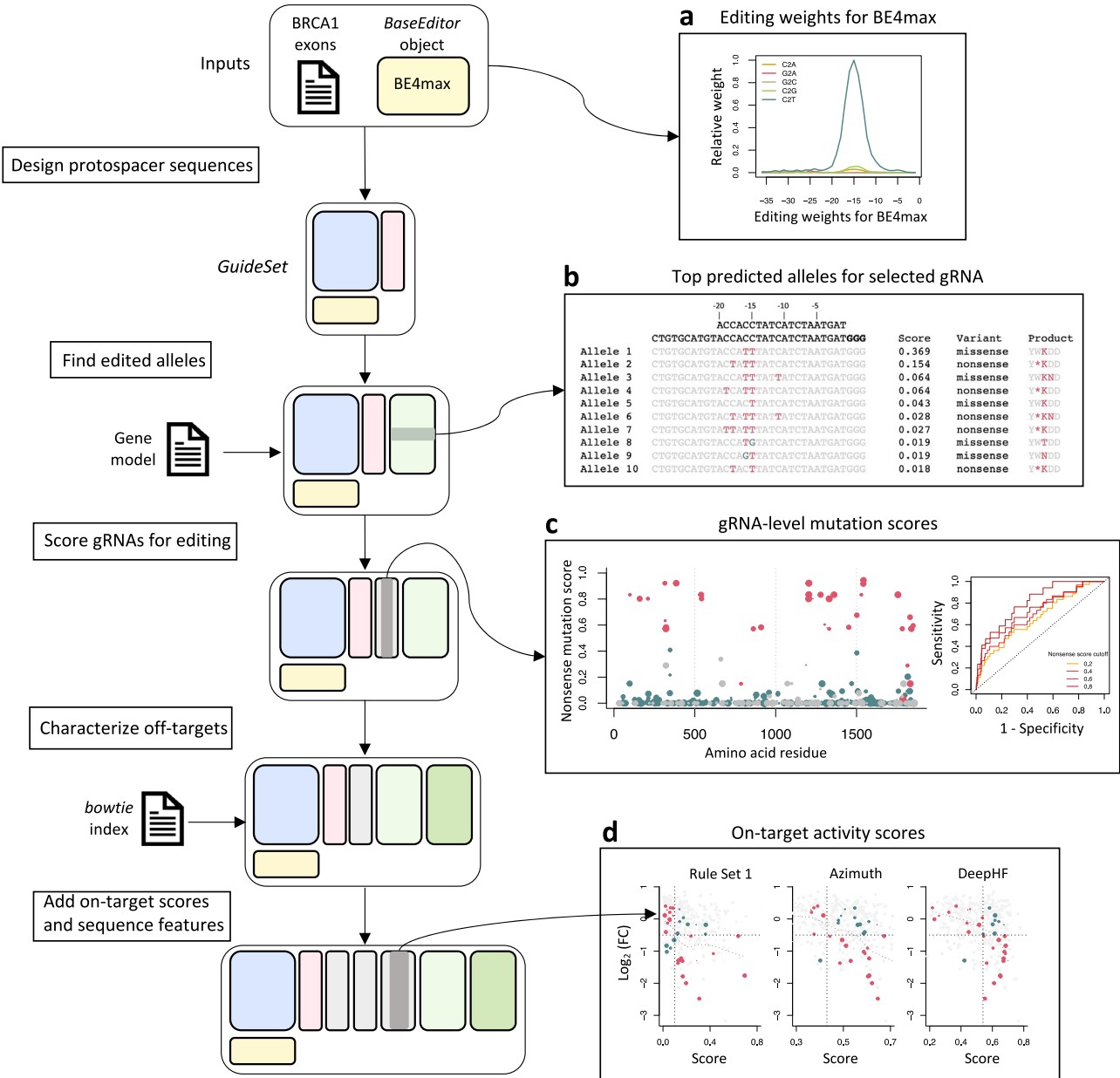

**Fig. 4 | *crisprDesign* workflow to design gRNAs tiling *BRCA1* using the base editor BE4max.** On the left: schematic showing the major steps involved in designing BE4max gRNAs targeting *BRCA1*. Two inputs are required: DNA sequences of *BRCA1* exons and a *BaseEditor* object from *crisprBase*. **a** Editing weights for the BE4max base editor from *crisprBase*. **b** 10 top predicted edited alleles for one selected gRNA as returned by *crisprDesign*. The wildtype allele and the protospacer sequence are positioned at the top of the first column, with the PAM sequence highlighted in bold. Edited nucleotides are highlighted in red (C to T) and blue (C to G). Editing scores, variant annotations, and protein product of the edited alleles are also shown. **c** On the left, gRNA-level nonsense mutation score as calculated by *crisprDesign*. Colors represent variant classification: nonsense in red, missense in blue, silent in gray. The size of the dot is proportional to the on-target

efficiency *DeepHF* score. On the right, ROC curves for classifying gRNA mutation type (nonsense or not) based on gRNA dropout from the *BRCA1* BE4max dataset (see Methods). Different thresholds of the nonsense score were used to label a gRNA as nonsense or not. **d** Relationships between gRNA dropout from the *BRCA1* BE4max dataset and several on-target activity scores. gRNAs that are not predicted to induce a nonsense mutation are colored in gray, and the size of the dots is proportional to the magnitude of the mutation score. The horizontal dotted lines at -0.5 represent a cutoff to classify a gRNA as active or not. For each method, a score cutoff was determined to classify active versus non-active gRNAs (vertical dotted line). Red and blue dots correspond to gRNAs that are correctly and incorrectly classified, respectively. Source data are provided as a Source Data file.

performed in HEK 293 cells with gRNA libraries tiling the human genes *CD46* and *CD55*[47]. Since both genes encode for cell-surface proteins, the authors used fluorescence-activated cell sorting (FACS) to sort cells with high and low expression. Their data can be used to investigate gRNA knockdown efficacy based on the change in relative abundance of high- and low-expressing cells for each targeted gene (see Methods).

We first extracted mRNA sequences of both genes using the function *getMrnaSequence* from *crisprDesign*. The mRNA sequences,

together with the *CrisprNuclease* object CasRx from *crisprBase*, served as inputs to create a *Guideset*. Next, we predicted on-target activity of the gRNAs using our implementation of the CasRx-RF method[47] available in *crisprScore* (see "Methods"). The normalized LFCs in the screen correlate well with the CasRx-RF score (Fig. 5a). We then added a transcript annotation to each gRNA using an Ensembl *TxDb* object as input. This adds a list of targeted isoforms to each gRNA, as well as transcript context (CDS, 5′UTR, or 3′UTR). We observed in the screen

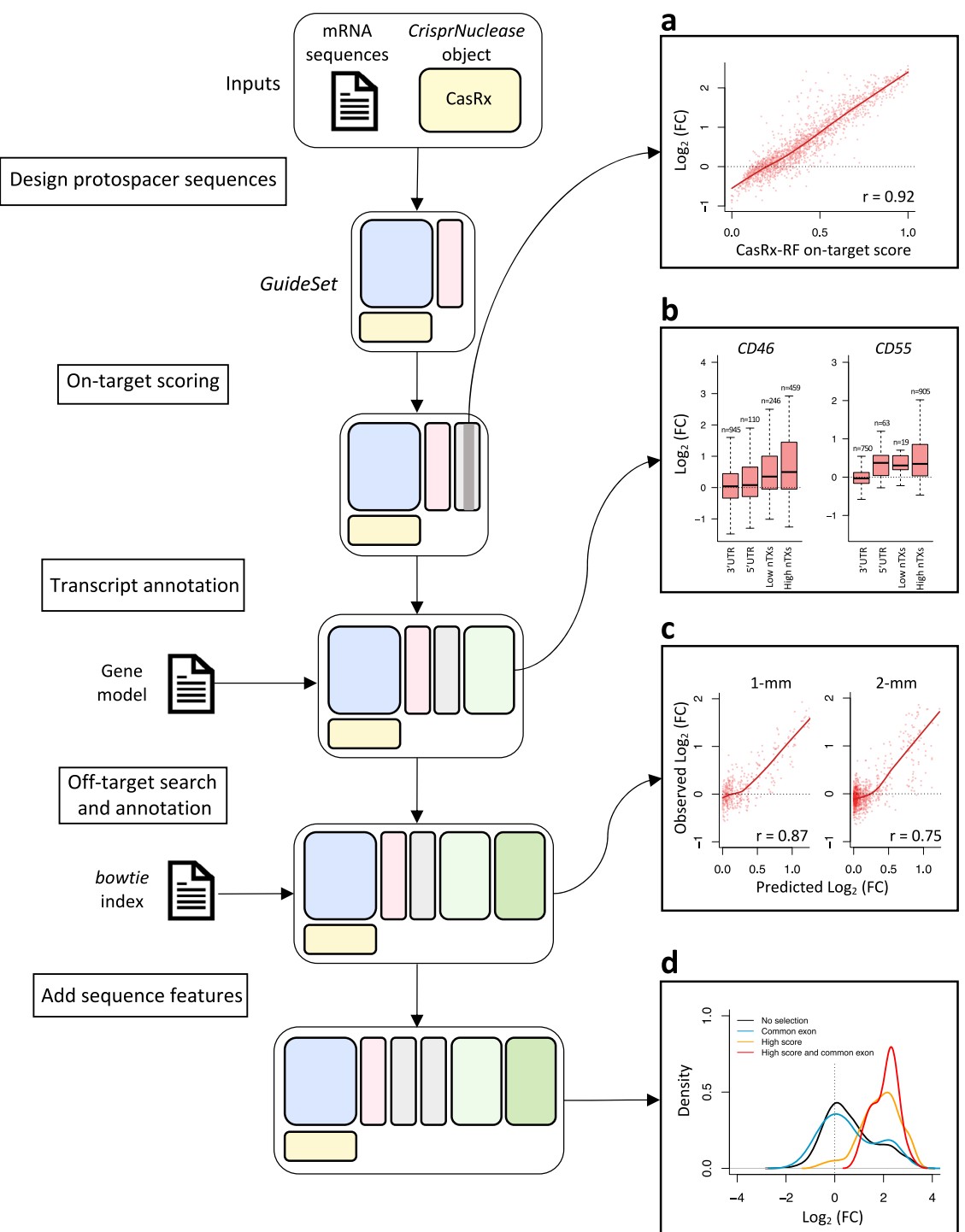

**Fig. 5 | *crisprDesign* workflow to design gRNAs tiling *CD55* and *CD46* using CasRx.** On the left: schematic showing the major steps involved in designing CasRx gRNAs targeting *CD55* and *CD46*. Two inputs are required: mRNA sequences of *CD55* and *CD46* and a *CrisprNuclease* object from *crisprBase*. **a** Relationship between on-target CasRx-RF score calculated in *crisprScore* and LFCs from the pooled FACS tiling CasRx screening data (see Methods). A higher LFC indicates higher gRNA activity. **b** Relationship between LFCs from the CasRx screening data and gRNA context for *CD46* and *CD55*: gRNAs targeting 5′ UTR and 3′ UTR for the canonical transcript, and guides targeting a low and high number of isoforms for each of the genes. gRNAs targeting more isoforms show higher enrichment in the screening data. The boxes represent the 25–75% interquartile ranges (IQR), and the central lines represent the median values. The whiskers extend 1.5 times the IQR from the median value. The number of data points for each boxplot is specified above the whiskers. The full isoform annotation is stored in the *GuideSet* objects. **c** Left: relationship between observed LFCs of on-target gRNAs in the *CD55* screen and predicted LFCs of single-mismatch gRNAs using the off-target CFD-CasRx score implemented in *crisprScore* (see Methods). Right: same as left, but for double-mismatch gRNAs. **d** gRNAs selected in the *CD46* screen for high on-target activity (CasRx-RF score) and targeting a common exon across all protein-coding isoforms enrich for high gRNA activity. Source data are provided as a Source Data file.

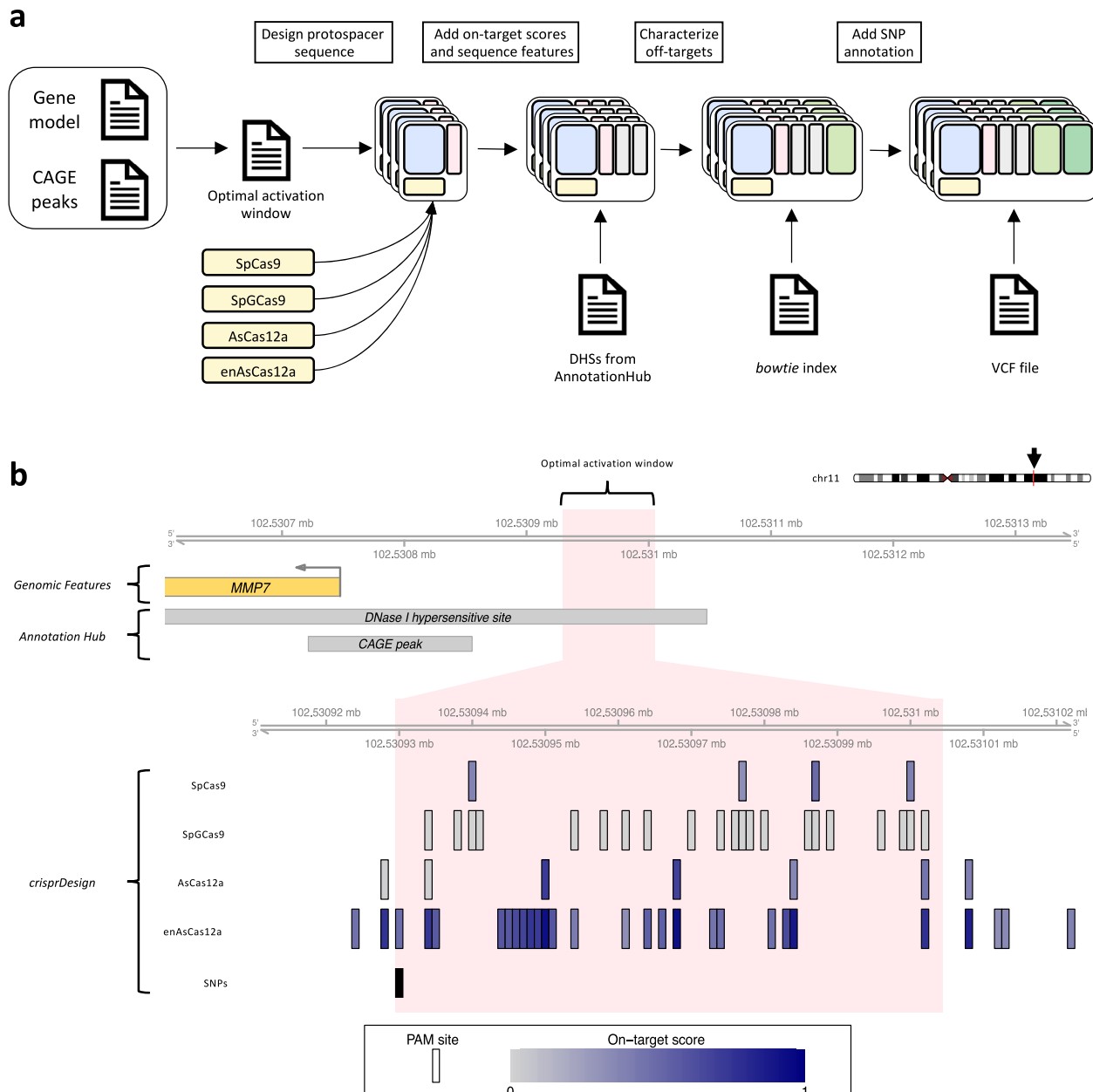

**Fig. 6 | Design of CRISPRa gRNAs for human gene *MMP7* for different CRISPR nucleases. a** Schematic showing the steps involved in designing CRISPRa gRNAs targeting the promoter region of *MMP7*. A gene model and a list of CAGE peaks are used to define the optimal window for gene activation. A *GuideSet* is created separately for each CRISPR nuclease. DNase I hypersensitive site (DHS) information is obtained from *AnnotationHub* and added to the gRNA annotation. **b** The top track shows the promoter region of human gene *MMP7* on chromosome 11, including part of the 5′ UTR of *MMP7* (yellow). The DHS and CAGE peak gray boxes were obtained using *AnnotationHub* (see Methods). The light pink region corresponds to the optimal region of activation, corresponding to a region [75,150]bp upstream of the 5′ end of the CAGE peak. For each of the four selected nucleases, all canonical PAM sites located within the optimal region are shown. PAM sites are colored by their on-target score: DeepHF for SpCas9, DeepCpf1 for AsCas12a, and enPAM+GB for enAsCas12a. No on-target scoring algorithm was available at time of publication for SpGCas9. The last track corresponds to common SNPs obtained from dbSNP151.

that gRNAs targeting a higher number of isoforms, and gRNAs located in CDS, lead to higher activity (Fig. 5b and Supplementary Fig. 4).

We performed an off-target search using *crisprBowtie* to the human transcriptome by providing a *Bowtie* index built on mRNA sequences. We extended the CFD off-target scoring algorithm implemented in *crisprScore* to work with CasRx by estimating mismatch tolerance weights on published GFP tiling screen data[47] (see "Methods"). The off-target CFD-CasRx score performs well at predicting gRNA activity of single-mismatch and double-mismatch gRNAs in the *CD55* screen (Fig. 5c and see "Methods").

Finally, we added sequence features, and ranked gRNAs for targeting *CD55* and *CD46* based on (1) high on-target score, (2) low number of off-targets, and (3) high number of targeted isoforms. If we select gRNAs that target a common exon and that have high on-target score, we enrich for highly active gRNAs in the screening data (Fig. 5d).

**Case study 3: Designing optimal gRNAs to activate *MMP7* using CRISPRa using different nucleases**

Designing gRNAs for either CRISPRa and CRISPRi applications requires additional considerations. This includes choosing an optimal target

region based on chromatin accessibility data and TSS data, and selecting gRNAs based on their positioning with respect to the TSS.

To demonstrate the utility of our ecosystem functionalities for CRISPRa and CRISPRi, we designed gRNAs for CRISPRa using the human gene *MMP7* as an example target (Fig. 6). CRISPRi is discussed at the end of this section. One CRISPRa-specific design consideration is the limited number of candidate gRNAs available for a given gene due to the narrow window of optimal activation. Engineered nucleases with less constrained PAM sequences can improve CRISPRa applicability by expanding the set of candidate gRNAs. To investigate this, we designed gRNAs for the promoter region of *MMP7* using four nucleases in parallel: SpCas9, AsCas12a, and the more PAM-flexible versions SpGCas9[21] and enAsCas12a[22].

The first step of the gRNA design was to specify the target region for *MMP7*. We used *AnnotationHub* to find CAGE peaks in the promoter region of *MMP7* to specify the TSS position. We used the CAGE data to identify TSSs instead of RefSeq or Ensembl as the former provides more accurate annotations for designing CRISPRi and CRISPRa gRNAs[79]. The 5′ end of the CAGE peak was used as the TSS to define the coordinates of the optimal window of activation. Based on a previous study[80], we defined the optimal window of activation to be between 75 and 150 nucleotides upstream of the TSS.

Next, we designed all possible gRNAs for the four nucleases using the *findSpacers* function in *crisprDesign*, and stored the gRNAs in four separate *GuideSet* containers. We annotated each *GuideSet* for overlap with DNase I hypersensitivity sites (DHS) from consolidated epigenomes from the Roadmap Epigenomics Project[81] using *AnnotationHub*. Open-chromatin regions are favorable for the binding of the catalytically inactive Cas9 (dCas9) used in both CRISPRa and CRISPRi[82,83]. We then added sequence features using *crisprDesign*, on-target scores using *crisprScore*, and off-target sites using *crisprBowtie* for each nuclease. Finally, we added overlapping SNPs information using the *addSNPAnnotation* function and using dbSNP151. The end-to-end workflow is presented in Fig. 6a.

The designed gRNAs are presented in Fig. 6b. With *crisprDesign*, it is straightforward to select candidate gRNAs in the most promising genomic regions - in this case, lying inside both the annotated DHS and the optimal activation window for MMP7. One can immediately appreciate that both nuclease variants (SpGCas9 and enAsCas12a) yield substantially more available gRNAs in the optimal window activation. In particular, enAsCas12a offers several gRNAs with high predicted on-target activity, making it a better candidate for gene activation of *MMP7*. One SNP was also found in the region of interest, and overlapping one gRNA for enAsCas12a that should be avoided. Altogether, our ecosystem provides an easy and comprehensive workflow to enable users to design optimal gRNAs for CRISPRa across nucleases.

Designing gRNAs for CRISPRi applications using *crisprDesign* is nearly identical, with the exception that the preferred target region for interference is located downstream of the TSS. The CRISPRai scoring algorithm[13], available through *crisprScore*, can be used to select optimal gRNAs for each TSS separately, taking into account both gRNA positioning and sequence content to maximize on-target inhibition. For both CRISPRa and CRISPRi, our gRNA design workflow is also applicable to non-coding regulatory elements, for instance long noncoding RNAs (lncRNAs)[84]. Overall, *crisprDesign* provides end-to-end functionalities that are well-suited for a large array CRISPRa and CRISPRi applications.

## Discussion

In this work, we introduced a suite of R packages to perform comprehensive end-to-end gRNA design for a multitude of CRISPR technologies and applications. Our ecosystem, named the *crisprVerse*, enables users to perform gRNA design for diverse nucleases such as PAM-free nucleases and RNA-targeting nucleases, and for several

applications beyond CRISPRko such as RNA and DNA base editing and CRISPRa/i. All design functionalities are available from a core package, *crisprDesign*. This eliminates the need to use multiple tools to obtain the necessary information for selecting optimal gRNAs, which is both time consuming and error prone. We demonstrated the diversity of our framework by applying it in three case studies involving different CRISPR technologies with their own specific design considerations.

We were able to show that creating rich gRNA annotations can help investigate gRNA variability and biases observed in experimental data generated from newer CRISPR technologies. To do so, we obtained public pooled screening data from two published studies, a tiling base editor screen of *BRCA1*, and a tiling CasRx screen of *CD46* and *CD55*, and show how some of the gRNA features derived from *crisprDesign* can explain some of the variability in gRNA activity observed in both screens. We also showed that our default gRNA ranking criteria implemented in *crisprDesign* yield optimal gRNAs by reanalyzing five genome-wide fitness screening datasets.

The modular architecture of the *crisprVerse* enables nucleases, base editors, scoring methods and annotations to be combined depending on the needs of the user. As a result, our design framework can easily adapt to new CRISPR technologies by swapping out the necessary components. For instance, a recent study has shown that the resolution of base editor screens can be greatly increased by combining existing base editors with PAM-extended Cas9 variants[85], while another study shows that RNA-targeting Cas13 nucleases can be combined with deaminases to form RNA base editors[28]. Both applications can be readily supported by our ecosystem without the need for further development.

Our ecosystem is completely implemented within the Bioconductor project, which provides robust and feature-rich data structures, high-quality documentation and workflows, and seamless interoperability between packages. Data structures defined in *crisprBase* can be reused to facilitate the analysis of CRISPR-based editing events in other packages, such as *ampliCan*[86], *GUIDEseq*[87] and *CrisprRVariants*[88]. *GuideSet* gRNA containers can be integrated with packages that provide analysis workflows for pooled screening data[89–91] to investigate biases and filter out undesirable gRNAs. Finally, the *crisprBowtie* and *crisprBwa* packages provide general functions that can be used to map any short sequences, including small-hairpin RNAs and short-interfering RNAs. We are continuously extending our suite of tools to make available the latest developments for gRNA design, such as prime editing[92] and combinatorial libraries[93].

## Methods

### Reference genomes, gene models, and genome indexes
The FASTA file for the human reference genome (GRCh38.p13 assembly) was obtained from UCSC to build *Bowtie* and *BWA* indexes via the *Rbowtie* (v1.37)[94] and *Rbwa* (v1.1) R packages, respectively. The packages use Bowtie v1.3 and BWA Release 0.7.17, respectively. The gene model used throughout the manuscript was obtained from Ensembl (release 104) using the R package *GenomicFeatures* (v1.49.6). Common SNPs were obtained from NCBI dbSNP build 151 (https://ftp.ncbi.nlm.nih.gov/snp/).

### CAGE peak and DNAse I hypersensitivity data
RIKEN/ENCODE CAGE peaks were obtained from *AnnotationHub* (v3.5) using accession number AH5084[95]. Genomic coordinates were lifted over from hg19 to hg38 using the R package *rtracklayer* (v1.57). DNAse I hypersensitive sites were obtained from *AnnotationHub* using accession number AH30743. The narrow DNase peaks were obtained using MACS2 on consolidated epigenomes from the Roadmap Epigenomics Project (`E116-DNase.macs2.narrowPeak.gz`)[81]. Genomic coordinates were lifted over from hg19 to hg38 using the R package *rtracklayer*.

## On-target scoring

We implemented several commonly-used algorithms for Cas9, Cas12 and Cas13 nucleases in *crisprScore*. For predicting on-target activity of the wildtype SpCas9 nuclease, we implemented the popular Rule Set 1[7] and *Azimuth* algorithms[8] (iteration of the popular Rule Set 2 algorithm by the same authors), and the sequence-only Rule Set 3[74]. The package also provides the deep learning-based algorithms *DeepWT* and *DeepHF*, developed to predict cutting efficiency of the wildtype SpCas9 and SpCas9-High Fidelity (SpCas9-HF1) nucleases, respectively[9], and the *DeepSpCas9* algorithm[45]. We also included the *CRISPRscan* algorithm[12] for predicting on-target activity of SpCas9 gRNAs expressed from a T7 promoter, as well as the *CRISPRater* algorithm[48]. For the wildtype AsCas12a, *crisprScore* offers the deep-learning based prediction method DeepCpf1[10]. For the enhanced AsCas12a (enAsCas12a), *crisprScore* offers the *enPAM+GB* algorithm[22]. For CasRx (RfxCas13d), we adapted the code from a published random forest model[47]; we referred to the method as *CasRx-RF*.

For predicting gRNA activity for CRISPRa and CRISPRi, we implemented the prediction method used to design the commonly-used Weissman CRISPRa and CRISPRi v2 genome-wide libraries for human and mouse[13]. This method predicts CRISPRa (or CRISPRi) gRNA activity based on the distance to the transcription starting site (TSS), spacer sequence-derived features, as well as chromatin accessibility data and nucleosome positioning using DNase-Seq, MNase-Seq, and FAIRE-Seq data. The chromatin data in hg38 coordinates are available on Zenodo (https://doi.org/10.5281/zenodo.6716721).

The function *addCompositeScores* from *crisprDesign* creates an aggregate score from a specified list of on-target scoring methods. It takes the average of the specified scores after performing a rank transformation. More specifically, consider $s_{ij}$ to be the score value for gRNA *i* and method *j*. The composite score $S_i$ for gRNA *i* is

$$S_i = \frac{\sum_{j=1}^{N} \text{rank}(s_{ij})}{N} \quad (1)$$

where *N* is the total number of user-specified on-target scoring methods, and rank($s_{ij}$) is the ranked score within method *j*. Importantly, if the number of missing values varies across on-target scoring methods, we ensure that the scale of the rank-transformed values are comparable across methods by simply scaling the ranks so that highest ranked value is equal across all methods. Missing values are uncommon but can happen when designing gRNAs targeting custom sequences. Indeed, several scoring algorithms require nucleotide context around the protospacer sequences, and this is not possible for gRNAs located near the end of the user-provided custom sequences.

### On-target prediction of frameshift-causing indels using *Lindel*.

In *crisprScore*, we implemented *Lindel*[46], a logistic regression model that was trained to use local sequence context to predict the distribution of mutational outcomes for CRISPR/Cas9. The *Lindel* final score reported in *crisprScore* is the proportion of "frameshifting" indels, that is the frequency of indels predicted to introduce frameshift mutations. By chance, assuming a random distribution of indel lengths, gRNAs should have a frameshifting proportion of 0.66. A *Lindel* score higher than 0.66 indicates that a given gRNA is more likely to cause a frameshift mutation than by chance.

### Off-target scoring of individual off-targets

The exact formula that we use to calculate the CFD score in *crisprScore* is

$$\text{CFD} = \prod_{p \in M} w_p(x_{\text{RNA}}, x_{\text{DNA}}) \quad (2)$$

where *M* is the set of positions for which there is a mismatch between the gRNA spacer sequence and the off-target sequence. $w_p(x_{\text{RNA}}, x_{\text{DNA}})$ is an experimentally-derived mismatch tolerance weight at position *p* depending on the RNA nucleotide $x_{\text{RNA}}$ and the DNA nucleotide $x_{\text{DNA}}$[6].

The exact formula that we use to calculate the MIT score in *crisprScore* was obtained from the MIT design website (crispr.mit.edu):

$$\text{MIT} = \left(\prod_{p \in M} c_p\right) \times \frac{1}{\frac{L-d}{L} \times 4 + 1} \times \frac{1}{m^2} \quad (3)$$

where *M* is the set of positions for which there is a mismatch between the gRNA spacer sequence and the off-target sequence, $c_p$ is an experimentally-derived mismatch tolerance weight at position *p*, *d* is the average distance between mismatches, *m* is the total number of mismatches, and *L* is the spacer length. The spacer length used in the original publication is 19[57]. As the number of mismatches increases, the cutting likelihood decreases.

### Composite off-target score for gRNA specificity

To create a gRNA-level composite specificity score, individual off-target cutting scores are aggregated using the following inverse summation formula:

$$\text{Specificity} = \frac{1}{1 + \sum_{i=1}^{n} C_i} \quad (4)$$

where $C_i$ is the cutting likelihood score (either using the MIT or the CFD method) for the $i^{\text{th}}$ putative off-target. A higher composite score indicates higher specificity, which decreases with more off-targets and/or a greater likelihood of cleavage at each off-target. A gRNA with no putative off-targets have a composite score of 1. A gRNA with 2 on-targets, that is a gRNA targeting two genomic loci with perfect complementarity, will have a composite score of 0.5.

### Evolutionary conservation scores

The function *addConservationScores* in *crisprDesign* annotates gRNAs with evolutionary conservation scores. It requires bigWig files containing basewise conservation scores, which can be easily obtained from the UCSC genome browser database[96] at the following link: https://hgdownload.soe.ucsc.edu/downloads.html. The gRNA score is calculated as the average conservation score of a region centered around the predicted cut site of the gRNA. By default, the width of the region is 18 nucleotides, but can be changed by users. For our analysis of human protein-coding genes, we used the phyloP score from an alignment of 29 genome sequences to the human genome available at https://hgdownload.soe.ucsc.edu/goldenPath/hg38/phyloP30way/. Positive phyloP scores indicate conserved regions, while negative scores indicate evolution faster than expected under neutral drift.

### Base editing scoring

The behavior of a base editor can be quantified in a 3-dimensional array of editing probabilities. Let *p* be the genomic position relative to the PAM site; let $nuc_u$ be the original nucleotide; and let $nuc_e$ be the edited nucleotide. Denote $q(p, nuc_u, nuc_e)$ as the probability that $nuc_u$ is edited to $nuc_e$ at position *p*. Experimental editing weights can be used, possibly after some adequate transformation, to obtain those probabilities.

To score the likelihood of each edited allele, we assume independence of editing events with respect to nucleotide position. Specifically, consider a wildtype allele $U = (u_{p_1}, u_{p_2}, ..., u_{p_n})$ and an edited allele $V = (v_{p_1}, v_{p_2}, ..., v_{p_n})$, where $u_{p_i}$ and $v_{p_i}$ are the nucleotides at position $_{p_i}$ relative to the PAM site for the wildtype and edited allele, respectively. The parameter *n* is chosen by the user, and should be large enough so that all nucleotides within the editing window of the chosen base editor are represented. We calculate the editing score for

the edited allele $V$ (with respect to the wildtype allele $U$) as follows:

$$S(U,V) = \prod_{i=1}^{n} q(p_i, u_{p_i}, v_{p_j}) \qquad (5)$$

For a given edited allele $V$, we classify the functional consequence of editing as either a silent, missense, or nonsense mutation. We use $f(V)$ to label the mutation. In case an edited allele results in more than one mutation, we choose the most consequential mutation as the label (nonsense over missense, and missense over silent). For a given gRNA targeting the wildtype allele $U$, and the set of all possible edited alleles $V_j$, we calculate an aggregated score for each mutation type by summing the editing scores across alleles for each mutation type. For instance, the aggregated score for silent mutations is calculated as follows:

$$S_{silent}(U) = \sum_{j=1}^{N} S(U,V_j)\mathbf{1}(f(V_j) = \text{silent}) \qquad (6)$$

where $N$ is the total number of possible edited alleles $V_j$.

**Creation of major and minor allele human genomes.** We built major and minor allele genomes for the hg38 build using common SNPs from the dbSNP151 RefSNP database. The "common" category is based on germline origin and a minor allele frequency (MAF) of $>= 0.01$ in at least one major population, with at least two unrelated individuals having the minor allele. See the dbSNP website https://www.ncbi.nlm.nih.gov/variation/docs/human_variation_vcf/ for more information. We excluded indels, and only considered SNPs that have MAF greater than 1% in the 1000 Genomes Project population. We then injected major alleles and minor alleles into the reference genome hg38 sequence to create "major allele" and "minor allele" genomes, respectively. Both resulting genomes are provided as standard FASTA files. We generated *Bowtie* and *BWA* indexes for the two genomes. All results files are available on Zenodo (https://doi.org/10.5281/zenodo.6862556). The two allele genomes are also available from Bioconductor via their respective packages:

- `BSgenome.Hsapiens.UCSC.hg38.dbSNP151.major`[97]
- `BSgenome.Hsapiens.UCSC.hg38.dbSNP151.minor`[98]

**Base editing pooled screen data analysis**
Fitness screen data in the MelJuSo cell line using a gRNA library tiling *BRCA1* were obtained from the supplementary material of the original publication[77]. We normalized the raw counts by scaling by the total number of reads, and $\log_2$-transformed the data. We filtered out low-abundance gRNAs that were further than 3 standard deviations below the mean in the plasmid (pDNA) sample. From the later timepoint samples, we subtracted from the pDNA sample log counts to obtain LFCs, and averaged the LFCs across replicates. We filtered out gRNAs targeting multiple loci, and gRNAs with off-targets (with up to 2 mismatches) located in genes other than *BRCA1*.

**CasRx pooled screen data analysis**
CasRx FACS pooled screening data tiling *CD55*, *CD46* and *GFP*[47], including processed and normalized LFCs for each gRNA https://gitlab.com/sanjanalab/cas13. We redesigned all possible gRNAs targeting any of the isoforms of *CD55* and *CD46* using *crisprDesign*, and considered only gRNAs also present in the pooled screening data for downstream analyses. We annotated all gRNAs with gene information (Ensembl release 104) and obtained off-targets with up to 3 mismatches for all gRNAs using *crisprBowtie*. We obtained CasRx-RF on-target activity scores using *crisprScore*. The transcripts annotated as canonical by Ensembl (ENST00000367042 for *CD46*, and ENST00000367064 for *CD55*) were used to visualize LFCs.

For each gRNA, we quantified the abundance of its target gene by summing transcript per million (TPM) counts in HEK-293 cells for all transcripts targeted by the gRNA. Transcript-level RNA quantification for HEK-293 cells was obtained from the Protein Atlas web portal https://www.proteinatlas.org, on March 5 2022. Data are based on The Human Protein Atlas version 21.0 and Ensembl version 103 . We averaged TPM counts across the two replicates.

We used the single-mismatch (SM) gRNA constructs from the GFP tiling screen to estimate position-dependent probabilities of mismatch tolerance by the CasRx nuclease. To do so, we first calculated differences in LFC ($\Delta$LFC) between SM gRNAs and their corresponding perfect-match (PM) gRNAs. We then fitted a LOESS curve with respect to the nucleotide position to obtain an average $\Delta$LFC at each spacer position (Supplementary Fig. 5a). We transformed the LOESS fitted values to a scale between 0 and 1 to represent them as percentages of activity with respect to the median activity of the PM gRNAs tiling GFP (Supplementary Fig. 5b). Given the sparsity of the data, specifying a nucleotide-specific weight at each position was not possible. We adapted in *crisprScore* the CFD off-targeting scoring method to CasRx by using those probabilities as scoring weights. The corresponding scoring algorithm is named CFD-CasRx.

To evaluate the performance of the CFD-CasRx score on an independent dataset, we calculated CFD-CasRx off-target scores on all SM and double-mismatch (DM) gRNAs included in the *CD55* tiling screen. To predict LFCs of the DM gRNAs, we multiplied their respective PM gRNA LFCs with the CFD-CasRx on-target scores.

**Evaluation of the off-target alignment methods within *crisprDesign***
For comparing runtimes of the off-target alignment methods, the following sets of gRNAs were chosen: (1) gRNAs targeting the coding sequence of *KRAS*, for a total of 52 gRNAs; (2) gRNAs targeting the coding sequence of *EGFR*, for a total of 645 gRNAs, and (3) gRNAs targeting the coding sequence of *ZNF101*, for a total of 152 gRNAs. The *KRAS* and *EGFR* cases represent small- and medium-sized sets of gRNAs. For *ZNF101*, a few gRNAs overlap a repeat element, and therefore have a high number of on- and off-targets. Alignment was performed to the GRCh38.p13 genome. The *Bowtie* and *Biostrings* alignment methods were evaluated using 0 to 3 mismatches, and the BWA alignment methods were evaluated using 0 to 5 mismatches. Run times were collected on a Macbook Pro with an Intel Core i7 CPU (2.6GHz, 6 cores, 16 GB memory).

**Comparison of off-target alignments across tools**
We compared computing times for designing SpCas9 gRNAs and performing a genome-wide off-target search for the following tools: *CCTop*, *CHOPCHOP*, *multicrispr*, *FlashFry*, and *crisprDesign*. The following tools were excluded from the comparison: *CRISPick* as it does not provide a standalone software; *CRISPRseek* as we were not able to complete the search within a reasonable time; *Cas-Designer* due to its requirement for specialized software that we were not able to install on our machines; *E-CRISP* as it was not possible to run their command line interface on customs DNA sequences or exons.

To perform the comparison, we generated six random subsets of protein-coding exons located on chr1 with the following sizes:100, 200, 400, 800, 1600 and 3200 exons. Off-target alignment was performed against the human reference genome (GRCh38 build) using a maximum of 2 mismatches for all methods. Run times were collected on a Macbook Pro with an Intel Core i7 CPU (2.6GHz, 6 cores, 16 GB memory). For each tool, parameters optimized for speed were chosen based on available documentation. In particular, the following parameters were used. For *CCTop*, we used `-totalMM 2 -coreMM 2 -maxOT 100000`. For *CHOPCHOP*, we used: `-fasta -G HG38 -t WHOLE -v 2`. For *FlashFry*, we used: `-maximumOffTargets 100000 -forceLinear -maxMismatch 2`. For *multicrispr*, we used Bowtie with 2 mismatches,

with no on-target scoring. For *crisprDesign*, we used the function *addSpacerAlignmentsIterative* with the *Bowtie* and *BWA* aligners with 2 mismatches.

## Processing of genome-wide screen datasets

**Achilles dataset.** CRISPRko fitness screening gRNA-level LFCs from Project Achilles (22Q2 release) were downloaded from the DepMap portal https://depmap.org/portal/download/all/. Processed LFCs representing changes in gRNA abundances between the last time point of the fitness screen and the plasmid DNA were available for 957 human cell lines. We used previously-published reference lists of essential and non-essential genes for normalization[72]. In particular, for each cell line, we first centered LFCs using the median value of the set of non-essential genes, and then scaled LFCs using the median value of the set of essential genes. This enables normalized LFCs to be comparable across cell lines. For each gRNA, we then summarized gRNA activity by averaging LFCs across cell lines.

**Hart2015 dataset.** Processed data from a genome-wide screen performed in HCT116 cells using the Toronto Knockout v1 (TKOv1) library[99] were downloaded from http://tko.ccbr.utoronto.ca/. We computed LFCs between Day 18 and Day 0.

**Hart2017 dataset.** Processed data from a genome-wide screen performed in HAP1 cells using the Toronto Knockout v3 (TKOv3) library[100] were downloaded from http://tko.ccbr.utoronto.ca/. We obtained available LFCs between Day 18 and Day 0.

**Wang2015 dataset.** Processed data from a genome-wide screen performed in K562 cells[101] were obtained from the supplementary material of the original publication. LFCs were calculated between the final and initial timepoints.

**Tzelepis2016 dataset.** Processed LFCs from a genome-wide screen performed in HL60 cells[102] were obtained from the supplementary material of the original publication

LFCs for the Hart2015, Hart2017, Wang2015 and Tzelepis2016 were further standardized using the approach used for the Achilles dataset, with the same sets of non-essential and essential genes. For each dataset, gRNAs were mapped to the set of human protein-coding genes found in the Ensembl release 104, and unmapped gRNAs were filtered out. Given that gRNAs with multiple on- and off-targets can confound the analysis of fitness screens[14], we removed gRNAs that map to multiple loci in the GRCh38 genome, as well as gRNAs with 1- and 2-mismatch off-targets located in coding regions other than the intended target. The final numbers of gRNAs further considered for analysis are presented in Table 4.

## Default gRNA rankings implemented in *crisprDesign*

For each nuclease, we rank gRNAs based on several rounds of priority. For SpCas9, gRNAs with unique target sequences and without one- or two-mismatch off-targets located in coding regions are placed into the first round. Then, gRNAs with a small number of one- or two-mismatch off-targets (less than 5) are placed into the second round. Remaining gRNAs are placed into the third round. Finally, any gRNAs overlapping a common SNP (human only), containing a polyT stretch, or with extreme GC content (below 20% or above 80%) are placed into the fourth round. For CRISPRko applications, within each round of selection, gRNAs targeting the first 85% of the coding sequence of the canonical Ensembl isoform, as well as gRNAs targeting conserved regions (phyloP conservation score greater than 0), are prioritized first. gRNAs with the same priority are then ranked by a composite on-target activity rank to further prioritize active gRNAs. Based on the consistently reliable performance performance and generalization of the methods *DeepHF* and *DeepSpCas9*[9,45,103], the composite on-target

activity rank is calculated by taking the average rank across the *DeepHF* and *DeepSpCas9* scores. For CRISPRa and CRISPRi applications, the CRISPRai on-target score is used instead of the composite score.

The process is identical for enAsCas12a, with the exception that the *enPAM+GB* method is used as the composite score given that it is the only method available for the enAsCas12a nuclease. For CasRx, gRNAs targeting at least 75% of the isoforms of a given gene, with no one- or two-mismatch off-targets, are placed into the first round. gRNAs targeting at least 50% of the isoforms of a given gene, with no one- or two-mismatch off-targets, are placed into the second round, and remaining gRNAs are placed into the third round. Finally, any gRNAs containing a polyT stretch, or with extreme GC content (below 20% or above 80%) are placed into the fourth round. Within each round of selection, gRNAs are further ranked by the *CasRxRF* on-target score, using the canonical Ensembl isoform for scoring.

## Generation of gRNA rankings from other tools

In addition to *crisprDesign*, we designed and ranked SpCas9 gRNAs for all human protein-coding (Ensembl release 104) using four additional tools. For *CHOPCHOP* (v3), we used the command line interface (CLI) available at https://bitbucket.org/valenlab/chopchop with default parameters. For *CCTop* (v1.0.0), we used the CLI available at https://bitbucket.org/juanlmateo/cctop_standalone with default parameters. For *FlashFry* (v1.15), we used the CLI available at https://github.com/mckennalab/FlashFry with default parameters. For *CRISPick*, due to the lack of a CLI, we submitted batch query jobs through the portal https://portals.broadinstitute.org/gppx/crispick/public (accessed on July 27 2022) with default parameters for the Hsu (2013) tracrRNA sequence using the Rule Set 3.

## Criteria used to compare feature availability across gRNA design tools

The following gRNA design tools were used for comparison in Table 1: *multicrispr* (v1.7.0), *CRISPRseek* (v1.37.2), *CHOPCHOP* (v3), *CRISPOR* (website v5.01), *CCTop* (v1.0.0), *Guides* (v1.0), *Cas-Designer* (v3.0), *FlashFry* (v1.15), *E-CRISP* (v5.4) and *CRISPick* (no version, accessed on July 27 2022). The criteria listed below were used for assessing feature availability.

*Nuclease section*: a check mark indicates support for the corresponding nuclease, and *Limited* indicates that only a subset of custom nucleases are available. *Modalities* section: a check mark indicates that the software offers at least one specific functionality for that modality. *Target space section*: for the *Reference genomes* row, a check mark indicates that the software supports gRNA design against reference genomes; for this row, *Limited* indicates that the versions of the reference genomes are outdated. For the *Custom sequences* row, a check mark indicates that the software supports the design of gRNAs targeting custom DNA sequences.

The *Off-target aligner* section indicates which alignment methods are available in each tool. The *Off-target options* section describes which off-target alignment functionalities are implemented: genomic coordinates of the off-targets are available to the user (*Genomic coordinates* row), off-target alignment to custom sequences (*Custom sequences* row), concurrent off-target alignment to multiple organisms (*cross-reactivity* row), and alignment to major or minor allele genomes (*Minor/major alleles* row). The *On-target* and *Off-target* scoring sections indicate which scoring methods are implemented in the software.

The *Annotations* section indicates whether or not users have access to several annotations in the gRNA outputs. *Off-target annotation* refers to gene context annotation of the off-targets; *Isoform specification* refers to information about which gene isoforms are targeted by a given gRNA; *Reinitiation sites* refers to gRNAs annotated as being upstream of potential reinitiation sites; *Pfam domains* refers to information about which Pfam domains are targeted by a given gRNA; *SNP*

*annotation* refers to an annotation of gRNAs overlapping common SNPs; *TSS annotation* refers to whether or not gRNAs are annotated to fall into the promoter region of knows TSSs; *Conservation* refers to evolutionary conservation annotation.

The *Library design* section indicates which library design features are available in each of the tools. *Restriction sites* indicates whether or not gRNAs can be filtered for restriction sites of common enzymes. *PolyT signal* indicates if PolyT stretch filtering is available. *GC content* indicates filtering based on percentage GC content. *Hairpin loops* indicates filtering based on potential self-complementarity. *Paired gRNAs* indicates whether or not design of paired gRNAs is enabled. *Ranking* indicates if the software returns a gRNA rank for user selection.

### Figure generation

All figures were made in R (4.2.1), with the exception of the following figures that were made in Microsoft PowerPoint (v16.64): Fig. 1, Fig. 2a, b, and the workflow diagrams of Figs. 4–6. Figures 2c, 6b were made using the R package *Gviz* (v1.41.1). Figures 3–5, Supplementary Fig. 1, Supplementary Fig. 3, Supplementary Fig. 4 and Supplementary Fig. 5 were made using base plotting functions in R. Reproducible code to generate all figures can be found in our GitHub manuscript repository.

### Reporting summary

Further information on research design is available in the Nature Research Reporting Summary linked to this article.

### Data availability

We deposited reprocessed chromatin accessibility data in K562 cells[13] used by the CRISPRai on-target algorithm on Zenodo (https://doi.org/10.5281/zenodo.6716721). We deposited fasta files, *Bowtie* indexes and *BWA* indexes for the major and minor alleles of hg38 using dbSNP151 on Zenodo (https://doi.org/10.5281/zenodo.6862556). We pre-computed and fully annotated gRNAs for human and mouse protein-coding genes using *crisprDesign* for the following nucleases: SpCas9, enAsCas12a, and CasRx. Ensembl release 104 and Ensembl release 102 were used to define genes for human and mouse, respectively. Separate datasets were generated for the CRISPRko, CRISPRa, CRISPRi, and CRISPRkd modalities. All files are available on Zenodo (https://doi.org/10.5281/zenodo.7042164). CasRx FACS pooled screening data tiling CD55, CD46 and GFP[47] were obtained from https://gitlab.com/sanjanalab/cas13. Dropout screen data in the MelJuSo cell line using a gRNA library tiling BRCA1[77] were obtained from https://www.cell.com/cms/10.1016/j.cell.2021.01.012/attachment/98851720-ecfa-49fb-947b-6f4c8976cbc5/mmc2.xlsx. Common SNPs were obtained from NCBI dbSNP build 151 (https://ftp.ncbi.nlm.nih.gov/snp/). RIKEN/ENCODE CAGE peaks were obtained from AnnotationHub using accession number AH5084[95]. DNAse I hypersensitive sites were obtained from AnnotationHub using accession number AH30743[81]. Achilles screening data[1] was obtained from the DepMap portal (https://depmap.org/portal/download/all). Hart2015[99] and Hart2017[100] datasets were obtained from http://tko.ccbr.utoronto.ca/. Wang2015 dataset[101] was obtained from https://www.science.org/doi/10.1126/science.aac7041. Tzelepis2016 dataset[102] was obtained from https://pubmed.ncbi.nlm.nih.gov/27760321/. Source data are provided with this paper.

### Code availability

All crisprVerse packages are open-source and available on GitHub (Table 2). At time of publication, all packages were accepted at Bioconductor and available on the development branch of Bioconductor. Because of its size, the data package *crisprDesignData* is hosted on GitHub only. Reproducible code of all analyses can be found at https://github.com/crisprVerse/crisprVersePaper and are archived on Zenodo

(https://doi.org/10.5281/zenodo.7217670). A list of extensive tutorials can be found at https://github.com/crisprVerse/Tutorials and are archived on Zenodo (https://doi.org/10.5281/zenodo.7212557). They can also be found in the Supplementary Software file.

The analyses included in this paper were produced using the following package versions:

*crisprDesign* (v0.99.178, https://doi.org/10.5281/zenodo.7217534) *crisprScore* (v1.1.17, https://doi.org/10.5281/zenodo.7217539) *crisprScoreData* (v1.1.4, https://doi.org/10.5281/zenodo.7212547) *crisprBowtie* (v1.1.2, https://doi.org/10.5281/zenodo.7217536) *crisprBase* (v1.1.8, https://doi.org/10.5281/zenodo.7217535) *crisprVerse* (v0.99.11, https://doi.org/10.5281/zenodo.7217532) *crisprBwa* (v1.1.5, https://doi.org/10.5281/zenodo.7217555) *Rbwa* (v1.1.1, https://doi.org/10.5281/zenodo.7212545) *crisprDesignData* (v0.99.24, https://doi.org/10.5281/zenodo.7212549) *crisprViz* (v0.99.23, https://doi.org/10.5281/zenodo.7217540)

We also offer a Docker container encapsulating the latest crisprVerse ecosystem on our DockerHub page (https://hub.docker.com/repository/docker/fortin946/crisprverse. Documentation about the installation and usage of the container can be found at the following link: https://github.com/crisprVerse/Docker.

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

## Acknowledgements

The authors thank Benjamin Haley, Mike Costa, Amy Heidersbach, Kristel Dorighi, Scott Martin, Rena Yang, Allison Vuong, Oleg Mayba, Sandra Melo Carlos, and Russell Xie for sharing their expertise with us and guiding the development of our software ecosystem. We also thank William Forrest, Maggie Crow, Hector Corrada Bravo, Michael Lawrence, and Benjamin Haley for providing invaluable feedback on the manuscript and software. We thank Nitesh Turaga, Lori Shepherd, Marcel Ramos, Helena Crowell and Kayla Morrell who kindly and thoroughly reviewed our R packages as part of the Bioconductor submission process.

## Author contributions

J.P.F. led the software development and supervised the work. J.P.F. conceptualized and wrote the manuscript, with contributions and input from all authors. L.H. and J.P.F. developed the R packages, with contributions from P.P. and A.L. All authors read and approved the final manuscript.

## Competing interests

J.P.F. and A.L. declare that they are Genentech/Roche employees and declare that they hold Roche stocks. The remaining authors declare no competing interests.
