## [Peer Review File · Nature Communications]

Reviewers' Comments:

Reviewer #1:

Remarks to the Author:

Researchers should consider a variety of criteria while designing CRISPR guide RNAs, including location, the number of on/off-target alignments, and the presence of polyT and the enzyme cut site. Hoberecht et al. demonstrated a complete and all-encompassing R workflow for designing CRISPR guide RNA. The pipeline can provide important information collected from multiple algorithms developed previously for various Cas nucleases, which is improved from previous efforts such as CRISPRseek and multicrispr. Furthermore, utilizing basilisk, this pipeline used not only R libraries but also python packages, allowing for an integrated analysis on a single R code. However, apart from giving useful additional information, it is uncertain whether it gives a better list of gRNAs of known nucleases than existing databases or algorithms. Overall, this well-designed and improved pipeline would be useful for researchers looking for the guide RNAs of novel nucleases, but authors should address the concerns below before publication.

Major comments

1. The crisprDesign pipeline provided a complete set of data gathered from several methods that had not been included in earlier algorithms. However, in terms of the rank of gRNAs, it is unclear that how they improved from previous databases or pipelines such as CRISPick (Broad) and results from CRISPRseek and multicrispr. A benchmark between the ranks of the current pipeline and previous pipelines should be included in this paper
2. Precalculated datasets for model organisms such as Human and Mouse should be provided as plain text file (through supplementary files or figshare) For those who are not familiar with R-based programming.
3. The best practices (i.e. whole genome search starting from a new type of a Cas nuclease or for new organisms) should be provided on crisprDesign Github for users who want to identify gRNAs for their Cas proteins. Also, in the demo, gRNAs were aligned to chr 12 only (the chromosome where IQSEC3 is located) for off-target search. However, the off-target search should be done across whole genome, so users may misinterpret the step.
4. The MIT off-target score formula seems based on a 20-bp gRNA. Should it be changed if the gRNA is 23 bp long?

Minor comments

1. Is the link the details of each APIs? <https://rdrr.io/github/Jfortin1/crisprDesign/man/> . I think it is good idea that the github page provide the url.
2. In my opinion, it is better to combine the R libraries (crisprDesign, crisprBase, crisprBowtie, cripsrBwa, crisprScore) into one library since all of them were needed for the pipeline and eventually installed together.
3. The tutorial link is broken on the cripsrScore github page.
4. Please specify version of bowtie, bwa, and other software.
5. `guideSet <- addOnTargetScores(guideSet)` was unsuccessful with an error below on mac R 4.2.0
Error in checkForRemoteErrors(lapply(cl, recvResult)) : one node produced an error:
PyException_SetTraceback - dlsym(0x7fde7d45a1b0, PyException_SetTraceback): symbol not found

Reviewer #2:

Remarks to the Author:

The manuscript from Hobrecht et al introduces a R-based framework for gRNA design, with a core package named `crisprDesign` (I will refer to the complete framework as `crisprDesign`). The framework is intended to efficiently design gRNAs and annotations for given targets, and provides functionality to consider a multitude of CRISPR technologies (CRISPRko, CRISPRa, CRISPRi, CRISPRbe). `crisprDesign` is implemented in a modular fashion, which allows to incorporate several alignment methods, annotations as well as on- and off-target activity scores. In order to rate the functionality of the packages, the authors compare it to two published methods. Further, they demonstrate usability by showing three use cases, namely CRISPRbe for BRCA1 gene, tiling targeting for CD46/CD55, and CRISPRa for MMP7, for which they design gRNAs taking advantage from different enzymes and PAMs.

Major:

Unfortunately, I was not able to install the `crisprDesign` framework on a fresh R 4.2 as suggested, as well as on an older R version (I used docker containers to exclude interference with existing packages). While with an older R version I was able to install most sub packages directly from github as given in the documentation (up to `crisprdesign`), I finally ended with an error that told me to use R 4.2. When using 4.2, I had already a large number of package errors when trying to install, that I was not able to solve within an appropriate time frame. Further, the manuscript and demo indicates the package to be deployed via Bioconductor, however, I was either able to install the package from there (demo on git: first command indicates availability on bioconductor), nor I was able to find the package at the bioconductor web page. I strongly suggest to either provide a functional docker container or to describe users how to use a suitable conda environment to simplify installation process.

Along this line, `crisprDesign` is for sure an interesting framework for the community dealing with CRISPR based screens. However, most features introduced are already implemented by individual tools or packages in R or python. The authors argue that `crisprDesign` will now streamline all these individual packages and will provide an overarching interface, an idea that I highly appreciate. They claim that the usage of their tool will simplify workflows and will come with higher usability. However, when looking at git and trying to use it, the authors do not further follow their best argument. They did not take into account who is the audience of their framework. Such overarching "wrapper and bridging" frameworks are complex by design, and potential users need a lot of help to start using it, which means a detailed documentation and various examples are of high importance. It also means that such frameworks will just become important for the CRISPR community, if people that not necessarily are fulltime bioinformaticians, find an entry point. Therefore, I highly suggest to significantly improve the documentation, tool usage, and example page in order to make the framework attractive for its potential audience. The authors might think about using the git wiki pages or `readthedocs`. I feel <https://scanpy.readthedocs.io> is a good example, how such a framework documentation should look like.

In the introduction, the authors introduce four key qualities for an ideal gRNA design framework that I fully agree with. However, from the manuscript I cannot derive sufficient evidence that the new `crisprDesign` tool is the only tool that fulfills these qualities, and comparison to existing tools should include core parameters that reflect these key qualities (see also next comment regarding quality (4)). The authors claim to solve these issues now, but lack to compare their tool to most other tools in the field often used (just two tools in their comparison). In the introduction, the authors name a whole bunch of tools, please include at least the later developed and/or often used ones such as Flashfry, CHOPCHOP, CRISPOR and CCTOP. For example, the Cas-Designer and Cas-OFFinder and its webpage (`rgentools`) provides a lot of functionality in terms of Cas-PAM selection, supported organisms and result filtering, one would like to take into account for comparison. Other examples are E_CRISP, GUIDES and CRISPick.

Especially for the last listed key quality (4: large scale gRNA design) the manuscript completely lack any evidence `crisprDesign` can handle such tasks. In my understanding, large scale gRNA libraries go for thousands of targets (e.g. all protein coding first exons etc.), and a key feature for such gRNA design tasks, among others, is runtime. Even the authors provide an iterative bowtie and bwa package and a speed comparison for a small design task on single genes (Figure S1), I did not find a hint that `crisprDesign` can provide acceptable runtimes on large scale designs.

In line with the prior critic, it lacks a speed comparison to other tools that are specialized on large scale designs, such as FlashFry or multicrispr.

Visualization of gRNAs in context of genomic loci as e.g. nicely shown in figure 2c and 5b are of high importance to choose the "best" gRNAs for single/few loci experiments. As one of the key

features of the given framework is to join data from multiple sources (SNPs, various annotations), I suggest to add functionality to visualize all this data. As far as I got it from git, there is no plotting functionality given yet.

Minor:

The last part of the discussion is interesting to read, and brings up a lot of good ideas that would greatly improve the functionality and usability of `crisprDesign`, however, I do not feel such future developing plans at that extend should be part of a discussion section.

Reviewer #3:

Remarks to the Author:

General comments

`CRISPRdesign` and its associated software packages provide tools for the design of various CRISPR-based editors and gene expression regulators, with useful features such as access to many published scoring matrices and SNP aware design. While these functions are available in existing web-based or command line (including Bioconductor) software packages, the availability of these tools in uniform environment and access to a variety of different published scoring matrices may benefit users who choose to invest in a new software package.

Detailed comments

1. The text is well written and the figures are nicely constructed. However, the manuscript does not provide the most important information – clear examples of input commands/files and the resulting output. Such vignettes are required components of analysis software packages in Bioconductor and should be provided as a supplemental file for the publication. These vignettes should include examples of all the major functions described in the publication. An example of gRNA design for genome-wide screens could be limited to a single technology applied to a subset of genes (e.g. annotated DNA repair or cell cycle genes).
2. Related to this comment, the software used to generate images in the figures should be clearly indicated so that the reader can discern which were generated directly with the described software packages.
3. A little more information about included gRNA ranking systems would be helpful. It appears that many options are available, but it is not clear if the package provides default or recommended ranking options for specific applications. Modification of gRNA ranking should be described as one of the vignettes.
4. The authors may wish to consider a different name than `CRISPRdesign` given a related package, also written in R, with a nearly identical name (`crisprDesignR`, PMID: 32494309.)
5. It seems that `CRISPRdesign` is not available in Bioconductor yet.
6. There are no examples in the Vignettes of `crisprScore` for three on-target scoring methods listed in Table 1, i.e., `CRISPRai`, `CasRx-RF`, and PAM scoring.

Response to referees for

A comprehensive Bioconductor ecosystem for the design of CRISPR guide RNAs across nucleases and technologies

Authors: Luke Hoberecht, Pirunthan Perampalam, Aaron Lun, Jean-Philippe Fortin*

We thank the reviewers for their careful reading of our work and their thoughtful comments, which have led to a much improved manuscript. We have substantially revised the manuscript and have added several sections and figures in the Appendix to detail additional analyses. In particular, we added the following components to our manuscript:

- Comparison of gRNA design functionalities across 11 gRNA design tools.
- New section on gRNA rankings, and benchmarking across state-of-the-art tools using 5 public genome-wide screening datasets.
- Speed comparison of the off-target alignment methods.
- Unification of the R packages into an ecosystem called *crisprVerse*, and creation of a website: <https://github.com/crisprVerse>.
- Creation of two new R packages, accepted by Bioconductor: crisprVerse and crisprViz for easy loading/installation of the ecosystem, and gRNA visualization functionalities, respectively.
- Creation of 16 comprehensive tutorials, available in the supplementary material and here.
- Creation of a Docker image available on Docker Hub.

We've attached a revised copy of our manuscript with changes highlighted in a light purple color. We address below each of the reviewers' comments.

Reviewer 1

Researchers should consider a variety of criteria while designing CRISPR guide RNAs, including location, the number of on/off-target alignments, and the presence of polyT and the enzyme cut site. Hoberecht et al. demonstrated a complete and all-encompassing R workflow for designing CRISPR guide RNA. The pipeline can provide important information collected from multiple algorithms developed previously for various Cas nucleases, which is improved from previous efforts such as CRISPRseek and multicrispr. Furthermore, utilizing basilisk, this pipeline used not only R libraries but also python packages, allowing for an integrated analysis on a single R code. However, apart from giving useful additional information, it is uncertain whether it gives a better list of gRNAs of known nucleases than existing databases or algorithms. Overall, this well-designed and improved pipeline would be useful for researchers looking for the guide RNAs of novel nucleases, but authors should address the concerns below before publication.

Major comments:

1. The *crisprDesign* pipeline provides a complete set of data gathered from several methods that had not been included in earlier algorithms. However, in terms of the rank of gRNAs, it is unclear that how they improved from previous databases or pipelines such as *CRISPick* (Broad) and results from *CRISPRseek* and *multicrispr*. A benchmark between the ranks of the current pipeline and previous pipelines should be included in this paper.

Response: This is a great suggestion. We have now performed a comprehensive benchmarking of gRNA rankings for pipelines that do provide rankings: *CCTop*, *CHOPCHOP*, *CRISPick*, *FlashFry* and our software *crisprDesign*. We note that *CRISPRseek* and *multicrispr* do not provide gRNA rankings. We focused our comparison on the set of all human protein-coding genes, and using the SpCas9 nuclease given the large amount of data for this nuclease that allow us to perform benchmarking across datasets, cell lines and studies. We downloaded and reprocessed 5 large-scale CRISPRko dropout screening datasets, performed with independent gRNA libraries, to perform the benchmarking. The results from our benchmarking analysis support that *crisprDesign* generates gRNA rankings lead to more optimized gRNAs in comparison to existing pipelines, as evidenced by its high performance across 5 independent large-scale screening studies. We have now added the following new section in the Results together with a new figure (Figure 1 below) and a new table (Table 1 below):

gRNA ranking from *crisprDesign* returns optimized gRNAs

To complement gRNA annotation and assist in library design, *crisprDesign* provides a gRNA ranking function called *rankSpacers*. The function implements our recommended ranking parameters for the nucleases SpCas9, enCas12a, and CasRx, effectively enabling library design automation across targets. It is designed to optimize both on-target activity and minimize off-targeting effects, and includes the functional annotations described in the previous section. Details are provided in the Methods section.

We compared our default gRNA ranking procedure to other tools that provide gRNA rankings: *CHOPCHOP*, *CCTop*, *FlashFry* and *CRISPick*. To perform the evaluation, we designed and ranked SpCas9 gRNAs for all human protein-coding genes (Ensembl release 104) using each tool separately (see Methods). Next, we obtained and processed 5 human genome-wide fitness screen datasets from published studies (Table 1), each performed using a different gRNA library. For each dataset and gRNA, a LFC between later and earlier time point samples was calculated to quantify gRNA dropout over time.

gRNAs targeting essential genes are expected to drop out and can be used for benchmarking purposes. To investigate the relationship between gRNA activity and gRNA ranking, we considered for each gRNA library the subset of gRNAs targeting a common reference set of essential genes [Hart et al., 2014]. For each gene and tool, we identified the top 15 ranked gRNAs based on the tool-specific in silico ranking. In Figure 1a, we show the distributions of LFCs in the Hart2015 dataset based on two groups: red lines show the distributions of the top 15 ranked gRNAs across genes, and green lines show the distributions of remaining gRNAs. Top ranked gRNAs from *CRISPick* and *crisprDesign* show greater activity than lower ranked gRNAs, as indicated by a negative shift in the red distributions with respect to the green distributions.

We repeated the analysis for each dataset, and summarized the performance of the top ranked gRNAs at discriminating active gRNAs by calculating the difference in means between the green and red distributions (Δ LFC). Results are shown in Figure 1b. Higher Δ LFCs indicate better

Dataset	gRNA library	Cell Line	Number of gRNAs	Reference
Achilles	Avana	(many)	67,816	Meyers et al. [2017]
Hart2015	TKOv1	HCT116	164,576	Hart et al. [2015]
Hart2017	TKOv3	HAP1	81,967	Hart et al. [2017]
Wang2015	Sabatini	K562	166,855	Wang et al. [2015]
Tzelepis2016	Yusa	HL60	85,192	Tzelepis et al. [2016]

Table 1: **Genome-wide human CRISPRko screen datasets used for comparing SpCas9 gRNA rankings**

performance, and results indicate that both *CRISPick* and *crisprDesign* perform well across all datasets.

We also added the two following sections in Methods:

Processing of genome-wide screen datasets

Achilles dataset: CRISPRko fitness screening gRNA-level LFCs from Project Achilles (22Q2 release) were downloaded from the DepMap portal <https://depmap.org/portal/download/all/>. Processed LFCs representing changes in gRNA abundances between the last time point of the fitness screen and the plasmid DNA were available for 957 human cell lines. Reference lists of essential and non-essential genes were downloaded from Hart et al. [2014]. For each cell line, we first centered LFCs using the median value of the set of non-essential genes, and then scaled LFCs using the median value of the set of essential genes. This enables normalized LFCs to be comparable across cell lines. For each gRNA, we then summarized gRNA activity by averaging LFCs across cell lines.

Hart2015 dataset: Processed data from a genome-wide screen performed in HCT116 cells using the Toronto Knockout v1 (TKOv1) library [Hart et al., 2015] were downloaded from <http://tko.cibr.utoronto.ca/>. We computed LFCs between Day 18 and Day 0.

Hart2017 dataset: Processed data from a genome-wide screen performed in HAP1 cells using the Toronto Knockout v3 (TKOv3) library [Hart et al., 2017] were downloaded from <http://tko.cibr.utoronto.ca/>. We obtained available LFCs between Day 18 and Day 0.

Wang2015 dataset: Processed data from a genome-wide screen performed in K562 cells were obtained from the supplementary material of Wang et al. [2015]. LFCs were calculated between the final and initial timepoints.

Tzelepis2016 dataset: Processed LFCs from a genome-wide screen performed in HL60 cells were obtained from the supplementary material of Tzelepis et al. [2016].

LFCs for the Hart2015, Hart2017, Wang2015 and Tzelepis2016 were further standardized using the approach used for the Achilles dataset, with the same sets of non-essential and essential genes. For each dataset, gRNAs were mapped to the set of human protein-coding genes found in the Ensembl release 104, and unmapped gRNAs were filtered out. Given that gRNAs with multiple on- and off-targets can confound the analysis of fitness screens [Fortin et al., 2019], we removed gRNAs that map to multiple loci in the GRCh38 genome, as well as gRNAs with 1- and 2-mismatch off-targets located in coding regions other than the intended target. The final numbers of gRNAs further considered for analysis are presented in Table 1.

Figure 1: Comparison of CRISPRko Cas9 gRNA rankings for protein-coding human genes. We designed and ranked gRNAs targeting all protein-coding human genes (Ensembl release 104) using tools that provide gRNA rankings: *CCTop*, *CHOPCHOP*, *FlashFry*, *CRISPick* and *crisprDesign*. To compare gRNA ranking performance across tools, we obtained gRNA LFCs from 5 genome-wide CRISPRko fitness screening datasets, listed in Table 1. In these fitness screens, active gRNAs targeting essential genes are expected to drop out and show negative LFCs. To investigate the relationship between gRNA activity and gRNA ranking, we considered for each gRNA library the subset of gRNAs targeting a common reference set of essential genes [Hart et al., 2014]. For each gene and tool, we identified the top 15 ranked gRNAs based on the tool-specific in silico ranking. **a** LFC distributions in the Hart2015 dataset for gRNAs targeting essential genes (solid lines) and gRNAs targeting non-essential genes (dotted lines). Red lines show the distributions of the top 15 ranked gRNAs across genes, and green lines show the distributions of remaining gRNAs. For essential genes, top ranked gRNAs from *CRISPick* and *crisprDesign* show greater activity than lower ranked gRNAs (red distributions are negatively skewed). As expected, there are no differences for gRNAs targeting non-essential genes. **b** We repeated the analysis described in **a** for each dataset. We summarized the performance of the top ranked gRNAs by calculating the difference in means between the green and red distributions (Δ LFC), for essential genes only. A higher Δ LFC indicates better performance. For each method and dataset, a t-test was performed to quantify the difference in LFCs between the top ranked gRNAs and the remaining gRNAs. Corresponding p-values are reported above the bars (*: p-value < 0.05; **: p-value < 0.01; ***: p-value < 0.001.)

Default gRNA rankings implemented in *crisprDesign*

For each nuclease, we rank gRNAs based on several rounds of priority. For SpCas9, gRNAs with unique target sequences and without one- or two-mismatch off-targets located in coding regions are placed into the first round. Then, gRNAs with a small number of one- or two-mismatch off-targets (less than 5) are placed into the second round. Remaining gRNAs are placed into the third round. Finally, any gRNAs overlapping a common SNP (human only), containing a polyT stretch, or with extreme GC content (below 20% or above 80%) are placed into the fourth round. For CRISPRko applications, within each round of selection, gRNAs targeting the first 85% of the coding sequence of the canonical Ensembl isoform, as well as gRNAs targeting conserved regions (phyloP conservation score greater than 0), are prioritized first. gRNAs with the same priority are then ranked by a composite on-target activity rank to further prioritize active gRNAs. Based on the consistently reliable performance and generalization of the methods *DeepHF* and *DeepSpCas9* shown in Konstantakos et al. [2022], Wang et al. [2019], Kim et al. [2019], the composite on-target activity rank is calculated by taking the average rank across the *DeepHF* and *DeepSpCas9* scores. For CRISPRa and CRISPRi applications, the CRISPRai on-target score is used instead of the composite score.

The process is identical for enAsCas12a, with the exception that the *enPAM+GB* method is used as the composite score given that it is the only method available for the enAsCas12a nuclease. For CasRx, gRNAs targeting at least 75% of the isoforms of a given gene, with no one- or two-mismatch off-targets, are placed into the first round. gRNAs targeting at least 50% of the isoforms of a given gene, with no one- or two-mismatch off-targets, are placed into the second round, and remaining gRNAs are placed into the third round. Finally, any gRNAs containing a polyT stretch, or with extreme GC content (below 20% or above 80%) are placed into the fourth round. Within each round of selection, gRNAs are further ranked by the *CasRxRF* on-target score, using the canonical Ensembl isoform for scoring.

Generation of gRNA rankings from other tools

In addition to *crisprDesign*, we designed and ranked SpCas9 gRNAs for all human protein-coding (Ensembl release 104) using four additional tools. For *CHOPCHOP* (v3), we used the command line interface (CLI) available at <https://bitbucket.org/valenlab/chopchop> with default parameters. For *CCTop* (v1.0.0), we used the CLI available at https://bitbucket.org/juanlmateo/cctop_standalone with default parameters. For *FlashFry* (v1.15), we used the CLI available at <https://github.com/mckennalab/FlashFry> with default parameters. For *CRISPick*, due to the lack of a CLI, we submitted batch query jobs through the portal <https://portals.broadinstitute.org/gppx/crispick/public> (accessed on July 27 2022) with default parameters for the Hsu (2013) tracrRNA sequence using the Rule Set 3.

2. Pre-calculated datasets for model organisms such as Human and Mouse should be provided as plain text file (through supplementary files or figshare) For those who are not familiar with R-based programming.

Response: This is a great suggestion, and we have now deposited fully-annotated and ranked gRNAs to Zenodo (doi: 10.5281/zenodo.7042164). We used the following configurations, using our recommended parameters:

- Human, SpCas9 (3 files corresponding to CRISPRko, CRISPRa, and CRISPRi)

- Human, enAsCas12a (3 files corresponding to CRISPRko, CRISPRa, and CRISPRi)
- Human, CasRx (1 file corresponding to CRISPRkd)
- Mouse, SpCas9 (3 files corresponding to CRISPRko, CRISPRa, and CRISPRi)
- Mouse, enAsCas12a (3 files corresponding to CRISPRko, CRISPRa, and CRISPRi)
- Mouse, CasRx (1 file corresponding to CRISPRkd)

We used GRCh38 with Ensembl Release 104 for human, and GRCm38 with Ensembl Release 102 for mouse. We have now added the following section in Data Availability:

We precomputed and fully annotated gRNAs for human and mouse protein-coding genes using *crisprDesign* for the following nucleases: SpCas9, enAsCas12a, and CasRx. Ensembl release 104 and Ensembl release102. were used for human and mouse, respectively. Separate datasets were generated for the CRISPRko, CRISPRa, CRISPRi, and CRISPRkd modalities. All files are available on Zenodo (doi: 10.5281/zenodo.7042164)

3. The best practices (i.e. whole genome search starting from a new type of a Cas nuclease or for new organisms) should be provided on *crisprDesign* Github for users who want to identify gRNAs for their Cas proteins. Also, in the demo, gRNAs were aligned to chr12 only (the chromosome where IQSEC3 is located) for off-target search. However, the off-target search should be done across whole genome, so users may misinterpret the step.

Response: We completely agree with the reviewer, and we have now created a set of 17 reproducible and comprehensive tutorials so that *crisprVerse* users can generate their own gRNA database using our best practices: crisprVerse tutorials. We have also included PDF copies of the tutorials in the Supplementary Material. For the *crisprDesign vignette*, a peculiarity of Bioconductor is that packages (including running and regenerating vignettes) have to be built under 5 minutes, which limits what we can do in complex vignettes such as the one in *crisprDesign*. This motivated us to create comprehensive tutorials that are not restricted by time. We have now changed the text in the *crisprDesign* vignette to the following to avoid misinterpretation, and redirected the user to the tutorial for more details re. how to perform an off-target search:

For the sake of time here, we will search only for on- and off-targets located in the beginning of human chr12 where IQSEC3 is located. We note note that users should always perform a genome-wide search as shown in the CRISPRko design tutorial.

4. The MIT off-target score formula seems based on a 20-bp gRNA. Should it be changed if the gRNA is 23 bp long?

Response:

We thank the reviewer for pointing that out. While the MIT score has not been used for longer spacer sequences, their formula easily generalizes to other spacer lengths. We have now rewritten the formula to generalize for any spacer length:

The exact formula that we use to calculate the MIT score in *crisprScore* was obtained from the MIT design website (crispr.mit.edu):

$$\text{MIT} = \left(\prod_{p \in M} w_p \right) \times \frac{1}{\frac{L-d}{L} \times 4 + 1} \times \frac{1}{m^2} \quad (1)$$

where M is the set of positions for which there is a mismatch between the gRNA spacer sequence and the off-target sequence, w_p is an experimentally-derived mismatch tolerance weight at position p , d is the average distance between mismatches, m is the total number of mismatches, and L is the spacer length. The spacer length used in Hsu et al. [2013] is 19. As the number of mismatches increases, the cutting likelihood decreases.

Minor comments:

1. Is the link the details of each APIs? <https://rdrr.io/github/Jfortin1/crisprDesign/man/> . I think it is good idea that the GitHub page provide the url.

Response: This is a good idea. We have now added a link to the manual page of each package (generated by Bioconductor) to the corresponding package GitHub page. As an example, we have added the following text to the *crisprDesign* GitHub page:

The complete documentation for the package can be found here.

2. In my opinion, it is better to combine the R libraries (*crisprDesign*, *crisprBase*, *crisprBowtie*, *crisprBwa*, *crisprScore*) into one library since all of them were needed for the pipeline and eventually installed together.

Response: This is a great suggestion, and we have now renamed our ecosystem to be called “*crisprVerse*” (in reference to the popular tidyverse ecosystem). We have created a GitHub organization to better organize our packages (crisprVerse website) as well as a package called *crisprVerse* to easily install and load all necessary packages in one line, now accepted at Bioconductor and also available here.

3. The tutorial link is broken on the *crisprScore* GitHub page.

Response: Sorry about that – we have now fixed the link.

4. Please specify version of *Bowtie*, *BWA*, and other software.

Response: We have now added the following versions in the Methods section:

AnnotationHub (v3.5)

crisprDesign (v0.99.134)

crisprScore (v1.1.14)

crisprScoreData (v1.1.3)

crisprBowtie (v1.1.1)

crisprBase v(1.1.5)

crisprVerse v(0.99.8)

crisprBwa (v1.1.3)

Rbwa (v1.1.0)

crisprDesignData (0.99.17)

crisprViz (0.99.18)

GenomicFeatures (v1.49.6)
rtracklayer (v1.57)
CHOPCHOP (v3)
CCTop (v1.0.0)
FlashFry (v1.15)
CRISPick (accessed on July 27 2022)
Bowtie (v1.3)
BWA (Release 0.7.17)
multicrispr (v1.7.0)
CRISPRseek (v1.37.2)
CRISPOR (v5.01)
Guides (v1.0)
Cas-Designer (v3.0)
E-CRISP (v5.4)

5. `guideSet <- addOnTargetScores(guideSet)` was unsuccessful with an error below on mac R 4.2.0

```
Error in checkForRemoteErrors(lapply(cl, recvResult)) : one node produced an error: PyException SetTraceback
```

Response: We thank the reviewer for bringing this to our attention, we have now fixed the error, which was due to a recent change in the *reticulate* package.

Reviewer 2

The manuscript from Hobrecht et al introduces a R-based framework for gRNA design, with a core package named *crisprDesign* (I will refer to the complete framework as *crisprDesign*). The framework is intended to efficiently design gRNAs and annotations for given targets, and provides functionality to consider a multitude of CRISPR technologies (CRISPRko, CRISPRa, CRISPRi, CRISPRbe). *crisprDesign* is implemented in a modular fashion, which allows to incorporate several alignment methods, annotations as well as on- and off-target activity scores. In order to rate the functionality of the packages, the authors compare it to two published methods. Further, they demonstrate usability by showing three use cases, namely CRISPRbe for BRCA1 gene, tiling targeting for CD46/CD55, and CRISPRa for MMP7, for which they design gRNAs taking advantage from different enzymes and PAMs.

Major comments:

(1) Unfortunately, I was not able to install the *crisprDesign* framework on a fresh R 4.2 as suggested, as well as on an older R version (I used docker containers to exclude interference with existing packages). While with an older R version I was able to install most sub packages directly from GitHub as given in the documentation (up to *crisprDesign*), I finally ended with an error that told me to use R 4.2. When using 4.2, I had already a large number of package errors when trying to install, that I was not able to solve within an appropriate time frame.

Further, the manuscript and demo indicates the package to be deployed via Bioconductor, however, I was either able to install the package from there (demo on git: first command indicates availability on Bioconductor), nor I was able to find the package at the Bioconductor web page. I strongly suggest to either provide a functional Docker container or to describe users how to use a suitable conda environment to simplify installation process.

Response: We thank the reviewer for thoroughly testing the installation of our software and providing this valuable feedback. We also apologize for the installation failure and the lack of instructions. To address this, we have taken several steps that should substantially facilitate the installation and improve the user experience.

First, we created a GitHub organization called “*crisprVerse*” (in reference to the popular tidyverse ecosystem) that locates all of our ecosystem packages in one place, together with extensive documentation and tutorials: (crisprVerse website).

Second, we created a new R package called *crisprVerse* to easily install and load all necessary packages in a couple of command lines (link). The package is accepted at Bioconductor and available on its devel branch.

Third, the crisprVerse website landing page describes the few steps necessary to install the *crisprVerse* packages. In addition, each package landing page also now contains an updated installation section (see the crisprBase page as an example).

Finally, following the reviewer’s suggestion, we also created a ready-to-use Docker image containing all necessary components for using the ecosystem: R (v4.2.1), the *crisprVerse* R packages and their dependencies, the *ViennaRNA* and *RNAhybrid* binaries necessary for some of the scoring functions, as well as the conda environments needed for running all of the scoring algorithms from in *crisprScore*. The Docker image is hosted on Docker Hub in this repository. We’ve also added the Dockerfile and documentation to our *crisprVerse* website here.

We believe those additions will significantly lower the bar for installation and usability of our ecosystem.

(2) Along this line, *crisprDesign* is for sure an interesting framework for the community dealing with CRISPR based screens. However, most features introduced are already implemented by individual tools or packages in R or python. The authors argue that *crisprDesign* will now streamline all these individual packages and will provide an overarching interface, an idea that I highly appreciate. They claim that the usage of their tool will simplify workflows and will come with higher usability. However, when looking at git and trying to use it, the authors do not further follow their best argument. They did not take into account who is the audience of their framework. Such overarching wrapper and bridging frameworks are complex by design, and potential users need a lot of help to start using it, which means a detailed documentation and various examples are of high importance. It also means that such frameworks will just become important for the CRISPR community, if people that not necessarily are full-time bioinformaticians, find an entry point. Therefore, I highly suggest to significantly improve the documentation, tool usage, and example page in order to make the framework attractive for its potential audience. The authors might think about using the git wiki pages or readthedocs. I feel <https://scanpy.readthedocs.io> is a good example, how such a framework documentation should look like.

Response: We absolutely agree with the reviewer, and our improvements described in response to comment (1) above are also in line with this comment. The crisprVerse website should facilitate usage and documentation by centralizing all of the components in one location.

We also created a markdown vignette for each package and made them available on the package landing pages, see the crisprBase page as an example. Those vignettes are also available on the

corresponding Bioconductor pages: see <https://www.bioconductor.org/packages/devel/bioc/vignettes/crisprBase/inst/doc/crisprBase.html> for `crisprBase`, for instance. Those vignettes are meant to show examples of the main features for each of the packages. We've included all vignettes (8 pdf documents) in the Supplementary Material.

We also created a series of 17 extensive and reproducible tutorials that we host on the `crisprVerse` website: see crisprVerse tutorials. Those tutorials were designed so that users can follow line by line how to design gRNAs for common applications. We also included PDFs of those tutorials in the Supplementary Material. Our hope is that the vignettes and tutorials should make our ecosystem more accessible to a broader audience.

Following Reviewer 1's suggestion, we also generated and annotated all possible gRNAs targeting human and mouse protein-coding genes for the following configurations:

- Human, SpCas9 (3 files corresponding to CRISPRko, CRISPRa, and CRISPRi)
- Human, enAsCas12a (3 files corresponding to CRISPRko, CRISPRa, and CRISPRi)
- Human, CasRx (1 file corresponding to CRISPRkd)
- Mouse, SpCas9 (3 files corresponding to CRISPRko, CRISPRa, and CRISPRi)
- Mouse, enAsCas12a (3 files corresponding to CRISPRko, CRISPRa, and CRISPRi)
- Mouse, CasRx (1 file corresponding to CRISPRkd)

We used GRCh38 with Ensembl Release 104 for human, and GRCm38 with Ensembl Release 102 for mouse. We have now added the following section in Data Availability:

We precomputed and fully annotated gRNAs for human and mouse protein-coding genes using *crisprDesign* for the following nucleases: SpCas9, enAsCas12a, and CasRx. Ensembl release 104 and Ensembl release 102 were used for human and mouse, respectively. Separate datasets were generated for the CRISPRko, CRISPRa, CRISPRi, and CRISPRkd modalities. All files are available on Zenodo (doi: 10.5281/zenodo.7042164).

(3) In the introduction, the authors introduce four key qualities for an ideal gRNA design framework that I fully agree with. However, from the manuscript I cannot derive sufficient evidence that the new *crisprDesign* tool is the only tool that fulfills these qualities, and comparison to existing tools should include core parameters that reflect these key qualities (see also next comment regarding quality (4)). The authors claim to solve these issues now, but lack to compare their tool to most other tools in the field often used (just two tools in their comparison). In the introduction, the authors name a whole bunch of tools, please include at least the later developed and/or often used ones such as *Flashfry*, *CHOPCHOP*, *CRISPOR* and *CCTOP*. For example, the *Cas-Designer* and *Cas-OFFinder* and its webpage (*rgentools*) provides a lot of functionality in terms of Cas-PAM selection, supported organisms and result filtering, one would like to take into account for comparison. Other examples are *E-CRISP*, *GUIDES* and *CRISPick*.

Response: That was a real oversight from our part. While we knew that most tools lacked several important features in one area or the other (which motivated the development of our ecosystem), we had indeed not provided any evidence of it in our original manuscript. We have now performed a comprehensive survey of gRNA design functionalities across our tool and 10 popular tools, including the ones mentioned by the reviewer (see Table 2 on the next page, now included in the manuscript). We believe this new comparison considerably strengthens our manuscript as it shows that each existing tools lack several important features in one or many of the 4 qualities mentioned above.

We have now included the following section in Results:

Table 2 includes a comparison to ten commonly-used gRNA design softwares: *multicrispr* [Bhagwat et al., 2020], *CRISPRseek* [Zhu et al., 2014], *CHOPCHOP* [Montague et al., 2014], *CRISPOR* [Concordet and Haeussler, 2018], *CCTop* [Stemmer et al., 2015], *GUIDES* [Meier et al., 2017], *Cas-Designer* [Park et al., 2015], *FlashFry* [McKenna and Shendure, 2018], *E-CRISP* [Heigwer et al., 2014] and *CRISPick*; see the Methods section for a detailed description of the criteria used for benchmarking. While several of the features implemented in *crisprDesign* are also available in other tools, *crisprDesign* provides the most complete gRNA design solution across nucleases and modalities. Unlike *crisprDesign*, many of the other tools do not provide informative on- and off-target annotations, limiting their use for optimal gRNA selection. In the following sections, we describe each of the gRNA design components and functionalities that are available in *crisprDesign*.

as well as the following section in Methods:

Criteria used to compare feature availability across gRNA design tools

The following gRNA design tools were used for comparison in Table 2: *multicrispr* (v1.7.0), *CRISPRseek* (v1.37.2), *CHOPCHOP* (v3), *CRISPOR* (website v5.01), *CCTop* (v1.0.0), *Guides* (v1.0), *Cas-Designer* (v3.0), *FlashFry* (v1.15), *E-CRISP* (v5.4) and *CRISPick* (no version, accessed on July 27 2022). The criteria listed below were used for assessing feature availability.

Nuclease section: a check mark indicates support for the corresponding nuclease, and *Limited* indicates that only a subset of custom nucleases are available. *Modalities section*: a check mark indicates that the software offers at least one specific functionality for that modality. *Target space section*: for the *Reference genomes* row, a check mark indicates that the software supports gRNA design against reference genomes; for this row, *Limited* indicates that the versions of the reference genomes are outdated. For the *Custom sequences* row, a check mark indicates that the software supports the design of gRNAs targeting custom DNA sequences.

The *Off-target aligner* section indicates which alignment methods are available in each tool. The *Off-target options* section describes which off-target alignment functionalities are implemented: genomic coordinates of the off-targets are available to the user (*Genomic coordinates* row), off-target alignment to custom sequences (*Custom sequences* row), concurrent off-target alignment to multiple organisms (*cross-reactivity* row), and alignment to major or minor allele genomes (*Minor/major alleles* row). The *On-target* and *Off-target* scoring sections indicate which scoring methods are implemented in the software.

The *Annotations* section indicates whether or not users have access to several annotations in the gRNA outputs. *Off-target annotation* refers to gene context annotation of the off-targets; *Isoform specification* refers to information about which gene isoforms are targeted by a given gRNA; *Reinitiation sites* refers to gRNAs annotated as being upstream of potential reinitiation sites; *Pfam domains* refers to information about which Pfam domains are targeted by a given gRNA; *SNP annotation* refers to an annotation of gRNAs overlapping common SNPs; *TSS annotation* refers to whether or not gRNAs are annotated to fall into the promoter region of known TSSs; *Conservation* refers to evolutionary conservation annotation.

The *Library design* section indicates which library design features are available in each of the tools. *Restriction sites* indicates whether or not gRNAs can be filtered for restriction sites of common enzymes. *PolyT signal* indicates if PolyT stretch filtering is available. *GC content* indicates filtering based on percentage GC content. *Hairpin loops* indicates filtering based on potential

self-complementarity. *Paired gRNAs* indicates whether or not design of paired gRNAs is enabled. *Ranking* indicates if the software returns a gRNA rank for user selection.

(4) Especially for the last listed key quality (4: large scale gRNA design) the manuscript completely lack any evidence *crisprDesign* can handle such tasks. In my understanding, large scale gRNA libraries go for thousands of targets (e.g. all protein coding, first exon, etc.), and a key feature for such gRNA design tasks, among others, is runtime. Even the authors provide an iterative bowtie and bwa package and a speed comparison for a small design task on single genes (Figure S1), I did not find a hint that *crisprDesign* can provide acceptable runtimes on large scale designs. In line with the prior critic, it lacks a speed comparison to other tools that are specialized on large scale designs, such as *FlashFry* or *multicrispr*.

Response: We completely agree with the reviewer, and we have now performed a comparison to *FlashFry* and *multicrispr*, as well as *CHOPCHOP* and *CCTop*. We found that both the Flashfry and the crisprDesign bowtie iterative alignment methods perform well in comparison to other softwares. We have now added the following results section

Finally, we compared run times for designing SpCas9 gRNAs and performing a genome-wide off-target search for the following tools: *CCTop*, *CHOPCHOP*, *multicrispr*, *FlashFry*, and *crisprDesign*. Other tools were not included for reasons discussed in the Methods section. To perform the evaluation, we generated six random subsets of protein-coding exons located on chromosome 1 with the following sizes: 100, 200, 400, 800, 1600 and 3200 exons. For each tool and each subset, we ran the off-target alignment against the human reference genome (GRCh38 build) using a maximum of 2 mismatches. We included the alignment parameters used for each tool in the Methods section. Both *FlashFry* and the iterative *Bowtie* alignment implemented in *crisprDesign* show a substantial speed gain in comparison to other methods (Figure S2).

as well as the following text in the Methods section:

Comparison of off-target alignments across tools

We compared computing times for designing SpCas9 gRNAs and performing a genome-wide off-target search for the following tools: *CCTop*, *CHOPCHOP*, *multicrispr*, *FlashFry*, and *crisprDesign*. The following tools were excluded from the comparison: *CRISPick* as it does not provide a standalone software; *CRISPRseek* as we were not able to complete the search within a reasonable time; *Cas-Designer* due to its requirement for specialized software that we were not able to install on our machines; *E-CRISP* as it was not possible to run their command line interface on custom DNA sequences or exons.

To perform the comparison, we generated six random subsets of protein-coding exons located on chr1 with the following sizes:100, 200, 400, 800, 1600 and 3200 exons. Off-target alignment was performed against the human reference genome (GRCh38 build) using a maximum of 2 mismatches for all methods. Run times were collected on a Macbook Pro with an Intel Core i7 CPU (2.6GHz, 6 cores, 16 GB memory). For each tool, parameters optimized for speed were chosen based on available documentation. In particular, the following parameters were used. For *CC-Top*, we used `--totalMM 2 --coreMM 2 --maxOT 100000`. For *CHOPCHOP*, we used: `--fasta -G HG38 -t WHOLE -v 2`. For *FlashFry*, we used: `--maximumOffTargets 100000 --forceLinear --maxMismatch 2`. For *multicrispr*, we used Bowtie with 2 mismatches, with no on-target scoring. For *crisprDesign*, we used the function *addSpacerAlignmentsIterative* with the *Bowtie* and *BWA* aligners with 2 mismatches.

We also added a new Supplementary figure (see Figure S2 below).

	crisprDesign	multicrispr	CRISPRseek	CHOPCHOP	CRISPOR	CCTop	GUIDES	Cas-Designer	FlashFry	E-CRISP	CRISPick
Nuclease	DNA-targeting: Cas9	✓	✓	✓	✓	✓	✓	✓	✓	✓	✓
	DNA-targeting: Cas12	✓	✓	✓	✓	✓	✓	✓	✓	✓	✓
	DNA-targeting: Custom	✓	Limited	Limited	Limited	Limited		✓	Limited		Limited
	RNA-targeting: Cas13 Nickase	✓	✓	✓	✓	✓	✓	✓	✓	✓	✓
Modalities	CRISPRko	✓	✓	✓	✓	✓	✓	✓	✓	✓	✓
	CRISPRbe	✓	✓	✓	✓	✓	✓	✓	✓	✓	✓
	CRISPRa	✓	✓	✓	✓	✓	✓	✓	✓	✓	✓
	CRISPRi CRISPRkd CRISPRkd OPS	✓	✓	✓	✓	✓	✓	✓	✓	✓	✓
Prime editing	*	✓	✓	✓	✓	✓	✓	✓	✓	✓	✓
Target space	Reference genomes	✓	✓	✓	✓	✓	Limited	✓	✓	✓	✓
	Custom sequences	✓	✓	✓	✓	✓	✓	✓	✓	✓	✓
Off-target aligner	Bowtie	✓	✓	✓	✓	✓	✓	✓	FlashFry	✓	†
	BWA Other	✓	Biostrings	Biostrings	✓	✓	✓	Cas-OFFinder	FlashFry	✓	†
Off-target options	Genomic coordinates	✓	✓	✓	✓	✓	✓	✓	✓	✓	✓
	Custom sequences	✓	✓	✓	✓	✓	✓	✓	✓	✓	✓
	Cross-reactivity	✓	✓	✓	✓	✓	✓	✓	✓	✓	✓
	Minor/major alleles	✓	✓	✓	✓	✓	✓	✓	✓	✓	✓
On-target scoring	Rule Set 1	✓	✓	✓	✓	✓	✓	✓	✓	✓	✓
	Azimuth	✓	✓	✓	✓	✓	✓	✓	✓	✓	✓
	Rule Set 3	✓	✓	✓	✓	✓	✓	✓	✓	✓	✓
	CRISPRscan	✓	✓	✓	✓	✓	✓	✓	✓	✓	✓
	CRISPRater	✓	✓	✓	✓	✓	✓	✓	✓	✓	✓
	DeepCpfI	✓	✓	✓	✓	✓	✓	✓	✓	✓	✓
	DeepSpCas9	✓	✓	✓	✓	✓	✓	✓	✓	✓	✓
	DeepHF	✓	✓	✓	✓	✓	✓	✓	✓	✓	✓
	Lindel	✓	✓	✓	✓	✓	✓	✓	✓	✓	✓
	CRISPRai	✓	✓	✓	✓	✓	✓	✓	✓	✓	✓
	EnPAM+GB	✓	✓	✓	✓	✓	✓	✓	✓	✓	✓
	CasRx-RF	✓	✓	✓	✓	✓	✓	✓	✓	✓	✓
	PAM scoring	✓	✓	✓	✓	✓	✓	✓	✓	✓	✓
Off-target scoring	MIT	✓	✓	✓	✓	✓	✓	✓	✓	✓	✓
	CFD	✓	✓	✓	✓	✓	✓	✓	✓	✓	✓
	CasRx	✓	✓	✓	✓	✓	✓	✓	✓	✓	✓
Annotations	Off-target annotation	✓	✓	✓	✓	✓	✓	✓	✓	✓	✓
	Isoform specification	✓	✓	✓	✓	✓	✓	✓	✓	✓	✓
	Reinitiation sites	✓	✓	✓	✓	✓	✓	✓	✓	✓	✓
	Pfam domains	✓	✓	✓	✓	✓	✓	✓	✓	✓	✓
	SNP annotation	✓	✓	✓	✓	✓	✓	✓	✓	✓	✓
	TSS annotation	✓	✓	✓	✓	✓	✓	✓	✓	✓	✓
	Conservation	✓	✓	✓	✓	✓	✓	✓	✓	✓	✓
	Restriction sites	✓	✓	✓	✓	✓	✓	✓	✓	✓	✓
Library design	PolyT signal	✓	✓	✓	✓	✓	✓	✓	✓	✓	✓
	GC content	✓	✓	✓	✓	✓	✓	✓	✓	✓	✓
	Hairpin loops	✓	✓	✓	✓	✓	✓	✓	✓	✓	✓
	Paired gRNAs	✓	✓	✓	✓	✓	✓	✓	✓	✓	✓
Ranking	✓	✓	✓	✓	✓	✓	✓	✓	✓	✓	

Table 2: gRNA design functionalities implemented in *crispr-Verse* and commonly-used gRNA design tools. Check marks indicate which functionalities are present in each tool at time of publication. * In progress. † Information could not be found. See the Methods section for a detailed description of the criteria used for assessing feature availability.

Supplementary Figure S2: **Comparison of computing times for subsets of human protein-coding exons.** We compared computing times across tools to design gRNAs and perform a genome-wide off-target search in the human genome. Six random subsets of protein-coding exons located on chr1 were used to perform the comparison. The sizes of the subsets were 100, 200, 400, 800, 1600 and 3200 exons. The x-axis shows the total size in nucleotides of the DNA target space formed by each subset, and the y-axis shows computing times in seconds. Details about the alignment parameters for each method can be found in the Methods section.

(5) Visualization of gRNAs in context of genomic loci as e.g nicely shown in figure 2c and 5b are of high importance to choose the best gRNAs for single/few loci experiments. As one of the key features of the given framework is to join data from multiple sources (SNPs, various annotations), I suggest to add functionality to visualize all this data. As far as I got it from git, there is no plotting functionality given yet.

Response: This is a great suggestion, and we have now added a package called crisprViz to our ecosystem, which enables users to visualize gRNAs within a genomic tracks together with data coming from multiple sources. We have added the following sentences to the manuscript:

To help with the design of complex libraries, we developed the package *crisprViz* to visualize *GuideSet* objects. The package builds on the Bioconductor package *Gviz* [Hahne and Ivanek, 2016] to offer a flexible and integrated visualization of gRNAs along genomic coordinates. Users can visually inspect gRNAs on a genomic track with the option of adding annotation tracks such as transcript models, SNP annotations, repeat elements, and nucleotide sequences.

Minor comments:

(1) The last part of the discussion is interesting to read, and brings up a lot of good ideas that would greatly improve the functionality and usability of *crisprDesign*, however, I do not feel such future developing plans at that extend should be part of a discussion section.

Response: We agree with the reviewer, and we have now replaced the following paragraph

We are continuously extending our suite of tools to make available the latest developments for gRNA design. We are currently extending our work to include design for prime editing [Anzalone et al., 2019]. Prime editing requires the design of template sequences in addition to the usual gRNA spacer sequences. Prime editing tools are rapidly evolving, and our infrastructure provides a robust framework to facilitate the complex design of a variety of prime editing constructs. In addition, we will implement more convenience functions to streamline the design of complex gRNA libraries, such as multiplexing libraries that combine multiple gRNA sequences within one construct. Such libraries can be used to increase on-target efficiency [Replogle et al., 2020], or to simultaneously target pairs of loci in the genome [Han et al., 2017]. We are also committed in making available new scoring algorithms by implemented them in *crisprScore* as they become available. Finally, to make access to our rich ecosystem accessible to all users, we are working on developing an interactive graphical user interface (GUI) using *Shiny* [Chang et al., 2015]. The Shiny application will combine user-friendly characteristics of existing web-based tools with the advanced capabilities of our R-based ecosystem.

by the following sentence:

We are continuously extending our suite of tools to make available the latest developments for gRNA design, such as prime editing [Anzalone et al., 2019] and combinatorial libraries [Replogle et al., 2020]

Reviewer 3

crisprDesign and its associated software packages provide tools for the design of various CRISPR-based editors and gene expression regulators, with useful features such as access to many published scoring matrices and SNP aware design. While these functions are available in existing web-based or command line (including Bioconductor) software packages, the availability of these tools in uniform environment and access to a variety of different published scoring matrices may benefit users who choose to invest in a new software package.

General comments:

1. The text is well written and the figures are nicely constructed. However, the manuscript does not provide the most important information – clear examples of input commands/files and the resulting output. Such vignettes are required components of analysis software packages in Bioconductor and should be provided as a supplemental file for the publication. These vignettes should include examples of all the major functions described in the publication. An example of gRNA design for genome-wide screens could be limited to a single technology applied to a subset of genes (e.g. annotated DNA repair or cell cycle genes).

Response: We absolutely agree with the reviewer, and thank them for the suggestion. We have extensively improved on the documentation of our ecosystem (now called *crisprVerse*) by doing

the following. First, we created a GitHub organization with one landing page: crisprVerse website to centralize all packages and facilitate usage and documentation.

Second, we created a markdown vignette for each package and made them available on the package landing pages, see the crisprBase page as an example. Those vignettes are also available on the corresponding Bioconductor pages: see <https://www.bioconductor.org/packages/devel/bioc/vignettes/crisprBase/inst/doc/crisprBase.html> for crisprBase, for instance. We've included all vignettes (8 pdf documents) in the Supplementary Material.

Finally, because the Bioconductor build system has strict time limits, which limits the scope of the package vignettes, we developed 17 extensive and reproducible tutorials that we host on our crisprVerse GitHub page: crisprVerse tutorials. PDFs of those tutorials are also included in the Supplementary Material.

2. Related to this comment, the software used to generate images in the figures should be clearly indicated so that the reader can discern which were generated directly with the described software packages.

Response: We have now added the following subsection in the Sections method:

Figure generation

All figures were made in R (4.2.1), with the exception of the following figures that were made in Microsoft PowerPoint (v16.64): Figure 1, Figure 2a-b, and the workflow diagrams of Figure 4, Figure 5 and Figure 6. Figure 2c and Figure 6b were made using the R package Gviz (v1.41.1). Figure 3, Figure 4, Figure 5, Figure S1, Figure S2, Figure S3 and Figure S4 were made using base plotting functions in R. Reproducible code to generate all figures can be found in our GitHub manuscript repository.

3. A little more information about included gRNA ranking systems would be helpful. It appears that many options are available, but it is not clear if the package provides default or recommended ranking options for specific applications. Modification of gRNA ranking should be described as one of the vignettes.

Response: We thank the reviewer for the great idea. We have refactored the *rankSpacers* function in *crisprDesign* to provide default and recommended ranking criteria, now discussed in the gRNA ranking section of the *crisprDesign* vignette, as well as in the *crisprDesign* package documentation. We have also added the following subsection in the Methods section of the manuscript:

Default gRNA rankings implemented in *crisprDesign*

For each nuclease, we rank gRNAs based on several rounds of priority. For SpCas9, gRNAs with unique target sequences and without one- or two-mismatch off-targets located in coding regions are placed into the first round. Then, gRNAs with a small number of one- or two-mismatch off-targets (less than 5) are placed into the second round. Remaining gRNAs are placed into the third round. Finally, any gRNAs overlapping a common SNP (human only), containing a polyT stretch, or with extreme GC content (below 20% or above 80%) are placed into the fourth round. For CRISPRko applications, within each round of selection, gRNAs targeting the first 85% of the coding sequence of the canonical Ensembl isoform, as well as gRNAs targeting conserved regions (phyloP conservation score greater than 0), are prioritized first. gRNAs with the same priority

are then ranked by a composite on-target activity rank to further prioritize active gRNAs. Based on the consistently reliable performance and generalization of the methods *DeepHF* and *DeepSpCas9* shown in Konstantakos et al. [2022], Wang et al. [2019], Kim et al. [2019], the composite on-target activity rank is calculated by taking the average rank across the *DeepHF* and *DeepSpCas9* scores. For CRISPRa and CRISPRi applications, the CRISPRai on-target score is used instead of the composite score.

The process is identical for enAsCas12a, with the exception that the *enPAM+GB* method is used as the composite score given that it is the only method available for the enAsCas12a nuclease. For CasRx, gRNAs targeting at least 75% of the isoforms of a given gene, with no one- or two-mismatch off-targets, are placed into the first round. gRNAs targeting at least 50% of the isoforms of a given gene, with no one- or two-mismatch off-targets, are placed into the second round, and remaining gRNAs are placed into the third round. Finally, any gRNAs containing a polyT stretch, or with extreme GC content (below 20% or above 80%) are placed into the fourth round. Within each round of selection, gRNAs are further ranked by the *CasRxRF* on-target score, using the canonical Ensembl isoform for scoring.

4. The authors may wish to consider a different name than *crisprDesign* given a related package, also written in R, with a nearly identical name (*crisprDesignR*, PMID: 32494309.)

Response: We must admit that this is indeed a rather unfortunate situation. We had chosen the name of our package a few years ago and we were completely unaware of this other package. At this time, both the extensive use of our package within our organization and its availability on Bioconductor preclude us to change its name.

5. It seems that *crisprDesign* is not available in Bioconductor yet.

Response: We apologize for this; *crisprDesign* was being reviewed by the Bioconductor team. All packages, including *crisprDesign*, are now accepted and available on the Bioconductor Devel branch, and will be available on the Production branch after their next release cycle (November 2022).

6. There are no examples in the Vignettes of *crisprScore* for three on-target scoring methods listed in Table 1, i.e., CRISPRai, CasRx-RF, and PAM scoring.

Response: We thank the reviewer for pointing this out. We have now fixed the *crisprScore* vignette to include examples for all scoring methods.

References

- Andrew V Anzalone, Peyton B Randolph, Jessie R Davis, Alexander A Sousa, Luke W Koblan, Jonathan M Levy, Peter J Chen, Christopher Wilson, Gregory A Newby, Aditya Raguram, et al. Search-and-replace genome editing without double-strand breaks or donor dna. *Nature*, 576(7785):149–157, 2019.
- Aditya M Bhagwat, Johannes Graumann, Rene Wiegandt, Mette Bentsen, Jordan Welker, Carsten Kuenne, Jens Preussner, Thomas Braun, and Mario Looso. multicrispr: grna design for prime editing and parallel targeting of thousands of targets. *Life science alliance*, 3(11), 2020.
- Winston Chang, Joe Cheng, JJ Allaire, Yihui Xie, and Jonathan McPherson. Package ‘shiny’. See <http://citeseerx.ist.psu.edu/viewdoc/download>, 2015.
- Jean-Paul Concordet and Maximilian Haeussler. Crispor: intuitive guide selection for crispr/cas9 genome editing experiments and screens. *Nucleic acids research*, 46(W1):W242–W245, 2018.
- Jean-Philippe Fortin, Jenille Tan, Karen E Gascoigne, Peter M Haverty, William F Forrest, Michael R Costa, and Scott E Martin. Multiple-gene targeting and mismatch tolerance can confound analysis of genome-wide pooled crispr screens. *Genome biology*, 20(1):21, 2019.
- Florian Hahne and Robert Ivanek. Visualizing genomic data using gviz and bioconductor. In *Statistical genomics*, pages 335–351. Springer, 2016.
- Kyuhoo Han, Edwin E Jeng, Gaelen T Hess, David W Morgens, Amy Li, and Michael C Bassik. Synergistic drug combinations for cancer identified in a crispr screen for pairwise genetic interactions. *Nature biotechnology*, 35(5):463–474, 2017.
- Traver Hart, Kevin R Brown, Fabrice Sircoulomb, Robert Rottapel, and Jason Moffat. Measuring error rates in genomic perturbation screens: gold standards for human functional genomics. *Molecular systems biology*, 10(7):733, 2014.
- Traver Hart, Megha Chandrashekar, Michael Aregger, Zachary Steinhart, Kevin R Brown, Graham MacLeod, Monika Mis, Michal Zimmermann, Amelie Fradet-Turcotte, Song Sun, et al. High-resolution crispr screens reveal fitness genes and genotype-specific cancer liabilities. *Cell*, 163(6):1515–1526, 2015.
- Traver Hart, Amy Hin Yan Tong, Katie Chan, Jolanda Van Leeuwen, Ashwin Seetharaman, Michael Aregger, Megha Chandrashekar, Nicole Hustedt, Sahil Seth, Avery Noonan, et al. Evaluation and design of genome-wide crispr/spcas9 knockout screens. *G3: Genes, Genomes, Genetics*, 7(8):2719–2727, 2017.
- Florian Heigwer, Grainne Kerr, and Michael Boutros. E-crisp: fast crispr target site identification. *Nature methods*, 11(2):122–123, 2014.
- Patrick D Hsu, David A Scott, Joshua A Weinstein, F Ann Ran, Silvana Konermann, Vineeta Agarwala, Yinqing Li, Eli J Fine, Xuebing Wu, Ophir Shalem, et al. Dna targeting specificity of rna-guided cas9 nucleases. *Nature biotechnology*, 31(9):827, 2013.
- Hui Kwon Kim, Younggwang Kim, Sungtae Lee, Seonwoo Min, Jung Yoon Bae, Jae Woo Choi, Jinman Park, Dongmin Jung, Sungroh Yoon, and Hyongbum Henry Kim. Spcas9 activity prediction by deepspcas9, a deep learning-based model with high generalization performance. *Science advances*, 5(11):eaax9249, 2019.
- Vasileios Konstantakos, Anastasios Nentidis, Anastasia Krithara, and Georgios Paliouras. Crispr-cas9 grna efficiency prediction: an overview of predictive tools and the role of deep learning. *Nucleic Acids Research*, 50(7):3616–3637, 2022.

- Aaron McKenna and Jay Shendure. Flashfry: a fast and flexible tool for large-scale crispr target design. *BMC biology*, 16(1):1–6, 2018.
- Joshua A Meier, Feng Zhang, and Neville E Sanjana. Guides: sgrna design for loss-of-function screens. *Nature methods*, 14(9):831–832, 2017.
- Robin M Meyers, Jordan G Bryan, James M McFarland, Barbara A Weir, Ann E Sizemore, Han Xu, Neekesh V Dharia, Phillip G Montgomery, Glenn S Cowley, Sasha Pantel, et al. Computational correction of copy number effect improves specificity of crispr–cas9 essentiality screens in cancer cells. *Nature genetics*, 49(12):1779–1784, 2017.
- Tessa G Montague, José M Cruz, James A Gagnon, George M Church, and Eivind Valen. Chopchop: a crispr/cas9 and talen web tool for genome editing. *Nucleic acids research*, 42(W1):W401–W407, 2014.
- Jeongbin Park, Sangsu Bae, and Jin-Soo Kim. Cas-designer: a web-based tool for choice of crispr-cas9 target sites. *Bioinformatics*, 31(24):4014–4016, 2015.
- Joseph M Replogle, Thomas M Norman, Albert Xu, Jeffrey A Hussmann, Jin Chen, J Zachery Cogan, Elliott J Meer, Jessica M Terry, Daniel P Riordan, Niranjan Srinivas, et al. Combinatorial single-cell crispr screens by direct guide rna capture and targeted sequencing. *Nature biotechnology*, 38(8):954–961, 2020.
- Manuel Stemmer, Thomas Thumberger, Maria del Sol Keyer, Joachim Wittbrodt, and Juan L Mateo. Cctop: an intuitive, flexible and reliable crispr/cas9 target prediction tool. *PloS one*, 10(4):e0124633, 2015.
- Konstantinos Tzelepis, Hiroko Koike-Yusa, Etienne De Braekeleer, Yilong Li, Emmanouil Metzakopian, Oliver M Dovey, Annalisa Mupo, Vera Grinkevich, Meng Li, Milena Mazan, et al. A crispr dropout screen identifies genetic vulnerabilities and therapeutic targets in acute myeloid leukemia. *Cell reports*, 17(4):1193–1205, 2016.
- Daqi Wang, Chengdong Zhang, Bei Wang, Bin Li, Qiang Wang, Dong Liu, Hongyan Wang, Yan Zhou, Leming Shi, Feng Lan, et al. Optimized crispr guide rna design for two high-fidelity cas9 variants by deep learning. *Nature communications*, 10(1):1–14, 2019.
- Tim Wang, Kıvanç Birsoy, Nicholas W Hughes, Kevin M Krupczak, Yorick Post, Jenny J Wei, Eric S Lander, and David M Sabatini. Identification and characterization of essential genes in the human genome. *Science*, 350(6264):1096–1101, 2015.
- Lihua J Zhu, Benjamin R Holmes, Neil Aronin, and Michael H Brodsky. Crisprseek: a bioconductor package to identify target-specific guide rnas for crispr-cas9 genome-editing systems. *PloS one*, 9(9):e108424, 2014.

Reviewers' Comments:

Reviewer #1:

Remarks to the Author:

The authors have substantially revised the paper, and have addressed all raised concerns. Additionally, biologists who are not familiar with R/linux will benefit from the pre-computed results that have been deposited via Zenodo.

Reviewer #2:

Remarks to the Author:

The authors answered and addressed all my critiques adequately and the manuscript was substantially improved by the additional analysis, edits, and manuals newly introduced. I suggest the editor to publish the manuscript in NC

Reviewer #3:

Remarks to the Author:

In this revision, the authors have addressed a number of concerns raised by the reviewers. The packages, tutorials, and vignettes are now available in Bioconductor. However, the features and documentation may not convince a large user base to switch from existing tools.

Given the large number of overlapping web-based and command line tools, the motivation to invest in this tool may be limited.

Response to referees for
*The crisprVerse: a comprehensive Bioconductor ecosystem for the
design of CRISPR guide RNAs across nucleases and technologies*

Authors: Luke Hoberecht, Pirunthan Perampalam, Aaron TL Lun, Jean-Philippe Fortin*

Reviewer 1

“The authors have substantially revised the paper, and have addressed all raised concerns. Additionally, biologists who are not familiar with R/linux will benefit from the pre-computed results that have been deposited via Zenodo.”

Response: We thank the reviewer for their great suggestions and reviewing our manuscript for a second time.

Reviewer 2

“The authors answered and addressed all my critiques adequately and the manuscript was substantially improved by the additional analysis, edits, and manuals newly introduced. I suggest the editor to publish the manuscript in NC.”

Response: We thank the reviewer for their great suggestions and reviewing our manuscript for a second time.

Reviewer 3

“In this revision, the authors have addressed a number of concerns raised by the reviewers. The packages, tutorials, and vignettes are now available in Bioconductor. However, the features and documentation may not convince a large user base to switch from existing tools. Given the large number of overlapping web-based and command line tools, the motivation to invest in this tool may be limited.”

Response: Given the extensive array of features implemented in our software and absent in existing tools (see Table 1 in the manuscript), we believe our tools has competitive and unique advantages. In particular, users who need to design gRNAs for emerging nucleases (e.g. Cas13d) and CRISPR modalities (e.g. CRISPR base editing) have limited options with current tools, and therefore will benefit from using our ecosystem. In addition, we show that our gRNA annotation features can inform better gRNA selection; we benchmarked our Cas9 gRNA ranking algorithm for human genes against existing tools, and we found that our gRNA ranking leads to the best performance across all

benchmarking datasets. Finally, our open-source and modular platform allows the community to reuse our components into their own softwares and tools, alleviating the need for other researchers to reimplement from scratch a large number of routinely needed algorithms such as the Bowtie and BWA-based off-target search. For instance, our crisprScore standalone package offers maintained and consistent implementations of a large number of on- and off-target algorithms written in different languages, many of which are not readily available in other tools. This enables other software developers to integrate state-of-the-art scoring algorithms into their own ecosystem without the burden of maintaining the several underlying Python conda environments.